


# Blowing snow detection from ground-based ceilometers: application to East Antarctica

Alexandra Gossart[1], Niels Souverijns[1], Irina Valerievna Gorodetskaya[2,1], Stef Lhermitte[3,1], Jan Thérèse Maria Lenaerts[4,1,5], Jan Herbert Schween[6], Alexander Mangold[7], Quentin Laffineur[7], and Nicole Petra Marie van Lipzig[1]

[1]Department of Earth and Environmental Sciences, KU Leuven, Leuven, Belgium
[2]Centre for Environmental and Marine Sciences, Department of Physics, University of Aveiro, Aveiro, Portugal
[3]Department of Geosciences and Remote Sensing, Delft University of Technology, Delft, the Netherlands
[4]Institute for Marine and Atmospheric research Utrecht, Utrecht University, Utrecht, The Netherlands
[5]Departement of Atmospheric and Oceanic Sciences, University of Colorado, Boulder CO, USA
[6]Institute of Geophysics and Meteorology, Koeln University, Koeln, Germany
[7]Royal Meteorological Institute of Belgium, Brussels, Belgium

*Correspondence to:* Alexandra Gossart (alexandra.gossart@kuleuven.be)

**Abstract.** Blowing snow impacts Antarctic ice sheet surface mass balance by snow redistribution and sublimation. Yet, numerical models poorly represent blowing snow processes, while direct observations are limited in space and time. Satellite retrieval of blowing snow are hindered by clouds and only consider the strongest events. Here, we develop a blowing snow detection algorithm for ground-based remote sensing ceilometers in polar regions. Results show that 79 % of the detected events are in agreement with visual observations. The algorithm is capable to detect both blowing snow lifted from the ground and occurring during precipitation, which is an added value since most of the blowing snow occurs during synoptic events, often combined with precipitation. Our analysis of atmospheric meteorological variables during blowing snow shows that blowing snow occurrence strongly depends on fresh snow availability in addition to wind speed, while the threshold for snow particles to be lifted is commonly parametrized as a function of wind speed only. These results suggest that the effect of katabatics and wind speed might have been overestimated, and that fresh snow availability should be considered in determining the blowing snow onset.

## 1 Introduction

Understanding the Antarctic ice sheet (AIS) response to atmospheric and oceanic forcing is crucial given its large potential impact on sea level rise (Rignot and Thomas, 2002; Rignot and Jacobs, 2002; Rignot et al., 2011; Shepherd et al., 2012). AIS mass balance is governed by the difference between surface mass balance (SMB) and solid ice discharging into the ocean. Solid precipitation is the only source term for the SMB. Meltwater runoff and surface sublimation are processes removing mass at the surface of the AIS, as well as the sublimation of the suspended snow particles. A fourth process is the erosion or re-deposition of transported snow particles from one location to another (Takahashi et al., 1988) : Snow particles can be dislodged from the snow surface and picked up by high wind speeds, and lifted from the ground into the near-surface atmospheric


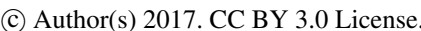


layer. This phenomenon occurs approximatively on 70 % of the Antarctic continent during winter (Palm et al., 2011) and snow is transported (1) by saltation, which is usually called "drifting snow" ( 0-0.3 m height, wind speeds from 2 to 5 $m \cdot s^{-1}$); (2) in suspension (layers up to 2 m high, wind speeds over 5 $m \cdot s^{-1}$), and (3) in blowing snow (wind speeds above 7 to 11 $m \cdot s^{-1}$, layers more than 2 m height (Frezzotti et al., 2004)).

Despite its importance, the role of blowing snow on local SMB and surface melt on the AIS is currently poorly quantified. If we consider the ice sheet in its whole, the contribution of blowing snow is rather small: around 0-6 % (Loewe, 1970; Déry and Yau, 2002; Lenaerts and van den Broeke, 2012). However, blowing snow is crucial for the local AIS SMB (Lenaerts and van den Broeke, 2012; Déry and Yau, 2002; Gallée et al. , 2001; Groot Zwaaftink et al., 2013) through the displacement and relocation of the snow particles (Déry and Tremblay, 2004) but also through sublimation (Takahashi et al., 1992; Thiery et al., 10   2012; Dai and Huang, 2014), an ablation process that contributes substantially to SMB above a threshold wind speed of 11 $m \cdot s^{-1}$ (Kodama et al., 1985) and more effective to remove mass than surface sublimation (van den Broeke et al., 2004). The combination of blowing snow sublimation and transport is estimated to remove from 50 to 80 % (van den Broeke et al., 2008; Scarchili et al., 2010; Frezzotti et al., 2004; van den Broeke, 1997) of the accumulated snow on coastal areas. Moreover, removal of the snow by the wind can locally lead to the formation of blue ice areas (Takahashi et al., 1988; Bintanja et al., 15   1995), which have a lower albedo and therefore enhance surface melt, and could affect ice shelf stability and collapse (Lenaerts et al., 2017). Blowing snow also plays a role in determining snow surface characteristics (Déry and Yau, 2002), affecting snow density and wind velocity threshold (Lenaerts and van den Broeke, 2012), and on surface energy balance (Lesins et al., 2009; Mahesh et al., 2003; Yamanouchi and Kawaguchi, 1985).

Many studies have focused on a minimum wind speed as threshold to dislodge snow particles, depending on the snow sur-
face properties (Budd et al., 1966). Schmidt (1980, 1982) explained that cohesion between snow particles requires higher wind speeds, or a higher impacting force of particles on the snow pack. In addition, the presence of liquid water in the snow and enhanced snow metamorphism with the higher atmospheric temperatures in the summer induce varying wind thresholds throughout the year (Bromwich, 1988; Li and Pomeroy, 1997).

Currently, simulations of the AIS SMB are highly uncertain since both blowing snow processes are poorly constrained and
probably lead to inconsistencies between the atmospheric modeled precipitations and the measured snow accumulation value (Frezzotti et al., 2004; Scarchili et al., 2010; Groot Zwaaftink et al., 2013; Gorodetskaya et al., 2015; van de Berg et al., 2005). In addition, strong blowing snow also hampers ground detection from satellites, and biases can be induced in efforts to study the Antarctic surface elevation due to the presence and radiative properties of blowing snow (Mahesh et al., 2002, 2003).

Efforts have been made to retrieve blowing snow from satellite data, but while it offers a large area coverage, the detection
is limited to clear-sky conditions and blowing snow layers thicker than 30 m (Palm et al., 2011), and make use of a wind threshold criterion. Moreover, ground validation remains essential to validate satellites measurements. A number of measurement campaigns have been organized in various regions of the AIS and used different types of devices (nets, mechanical traps and rocket traps, photoelectric and single-beam photoelectric sensors, and various studies have also worked with Flow-Capts or piezoelectric devices, (Leonard et al., 2011; Amory et al., 2015; Trouvilliez et al., 2015; Barral et al., 2015)). However,
custom-engineered sensors are rather expensive and scarce (Leonard et al., 2011), and both the remoteness of the continent and




the harshness of the climate are limitations to widespread use of these devices.

In this study we propose a new method to detect blowing snow by the use of ground-based remote sensing ceilometers. Ceilometers are robust cloud base height detection devices. Initially located in airports and designed to report visibility for pilots, the backscatter signal of these ground-based low-power lidars contain further information. These have been widely used
for scientific purposes regarding boundary layer investigation (Thomas, 2012; Marcowicz et al., 1997; Eresmaa et al., 2006; Heese et al., 2010): detection and vertical extent of aerosol layers below 5 km, mixing height layers (Haeffelin et al., 2012), as well as the detection of the early stage of radiation fog (Haeffelin et al., 2016). Several algorithms have been developed to detect cloud base height in specific areas, at the polar regions using the polar threshold algorithm (Van Tricht et al., 2014) or at temperate latitudes with the temporal height tracking algorithm (Martucci et al., 2010). Ceilometer networks are also
developed as a potential to cover larger regions (Illingworth et al., 2015). Over the Antarctic continent, the environmental conditions imply that research stations are usually equipped with robust instruments, that are able to withstand cold and difficult circumstances. Ceilometers can be operated autonomously and continuously in climatic conditions between -40 and +60 °C, up to 100 % relative humidity, and $50 \, \mathrm{m \cdot s^{-1}}$ (Vaisala User's guide, 2006). Compared to lidars, ceilometers have numerous advantages: e.g. eye-safe operation, low first range gate and relative low price, making it one the most abundant cloud detection
device on the ice sheets (Van Tricht et al., 2014; Wiegner et al., 2014).

The goal of this paper is to present a new methodology for blowing snow detection (BSD) using the ceilometer attenuated backscatter profile, and estimate the frequency of blowing snow at Neumayer III and Princess Elisabeth stations. Subsequently, we apply the BSD algorithm and investigate the near surface atmospheric changes during blowing snow, and we discuss blowing snow and the associated meteorological regimes. We conclude by examining the applicability of the BSD algorithm to
other Antarctic sites.

## 2   Instrumentation and location

### 2.1   Ceilometers

Ceilometers are rather simple and robust instruments. They consist of a single-wavelength, eye-safe active laser transmitter that emits pulses in the vertical direction, and an avalanche diode receiver that collects the pulse signal. The laser pulse
backscattered by molecules, aerosols, precipitation and cloud particles present in the atmosphere at height $z$, is detected by the ceilometer receiver. Typically, the backscatter intensity depends on the concentration or size of particles in the air, but the ceilometer receiver also detects noise induced by the device's electronics and the background light. The lidar equation enables to get the return signal strength from the emitted laser pulse (Münkel et al., 2006). As equation 1 displays:

$$\beta_{att}(a) = \beta(z) \cdot \tau^2(z) \tag{1}$$

the attenuated backscatter profile at the range $a$, $\beta_{att}$ ($\mathrm{sr^{-1} \cdot m^{-1}}$) is a product of the true backscatter coefficient $\beta$ at distance $z$, taking into account the two way attenuation of the lidar due to the transmittance of the atmosphere ($\tau^2$), and a height normalization is applied to the retrieved signal. This, to remove the excessive decrease in backscatter intensity in the presence





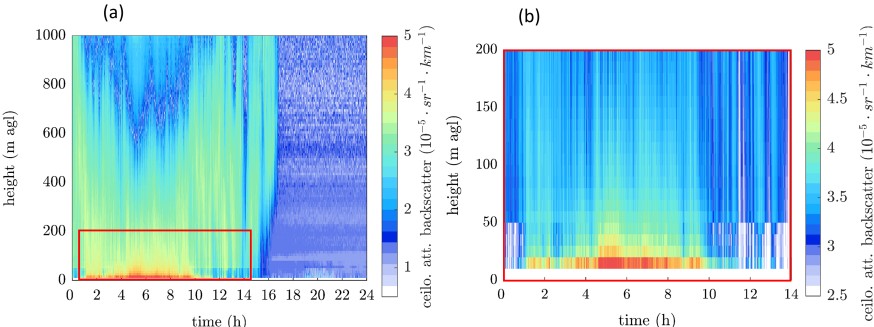

**Figure 1.** (a) Time (x-axis, in h)- height (y-axis, m agl) cross section of an attenuated backscatter profile for the CL-31 ceilometer at PE station, on April 24, 2016. The colour of the profile represents the intensity of the returned backscattered signal at a certain range bin. (b) Zoom onto blowing snow between 1:00 and 10:00 UTC, denoted by the red and yellow color in the range bins closest to the ground. The artefact discussed in section 3.1. is visible around 50 m.

of fog or precipitation between the instrument and the cloud base (Gorodetskaya et al., 2015). Finally, the detected signal is summed to a resolution of 15 s (to increase signal-to-noise ratio, Vaisala User's guide (2006)) at a spatial resolution of 10 m. The detected signal is reported at the centre of the 10 m range gate (i.e. for a signal measured between 50 and 60 m, the value of the range gate will be attributed to a height of 55 m (range bin 5)). The ceilometer measures continuously and the standard

output, $\beta_{att}$ is displayed in a time-height cross section (Fig. 1).

The cloud-base height is the standard ceilometer output determined from the backscattered signal: using the time delay between the launch of the pulse and its reception, and knowing the speed of displacement (the speed of light). Secondly, the instantaneous magnitude of the signal received by the diode provides information on the backscattering properties of the atmosphere, at determined heights. The only quantitative particle property that can be derived from the ceilometer measurements is

the attenuated backscatter intensity (Wiegner et al., 2014; Madonna et al., 2015). Other properties such as optical depth, size and density would require to know the lidar ratio. This is only possible if the ceilometer is calibrated, which is very challenging since the signal to noise ratio has to be large enough in the troposphere (Wiegner et al., 2014) and is not done in the present study.

## 2.2    The Cloud-Precipitation observatory at Princess Elisabeth station

The Princess Elisabeth (PE) station is located on the Utsteinen ridge in Dronning Maud Land (DML), East Antarctica (71 °57' S and 23 °21' E at 1392 m asl and 173 km inland, Fig. 2). A cloud and precipitation observatory was set up on the roof of the station (approx. 10 m above the ridge) during the summer season of 2009-2010 and is still operational under the Hydrant/Aerocloud project (www.aerocloud.be). The observatory contains an automatic weather station (AWS) and a set of ground-based remote sensing instruments: a Vaisala CL-31 ceilometer, a Heitronix infrared pyrometer and a Metek vertically

profiling precipitation radar, with a webcam for weather and instrument status monitoring. The observatory was designed to be



**Table 1.** Vaisala CL-31 ceilometer and CL-51 ceilometer specifications

| Type | CL31 | CL51 |
|------|------|------|
| range (m) | 10 - 7700 | 10 - 13500 |
| reporting resolution (m) | 10 | 10 |
| reporting cycle (s) | 2-120 | 16-120 |
| measurement interval (s) | 2 | 2 |
| reporting interval (s) | 15 | 15 |
| laser wavelength (nm) | $910 \pm 10$ at 25 °C | 910 |

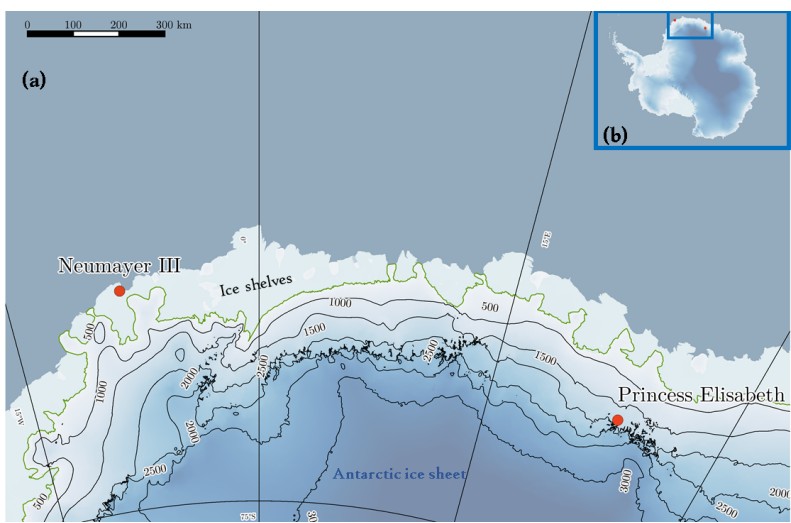

**Figure 2.** (a) Topographic map of the location of Neumayer and Princess Elisabeth stations in DML. The color intensity represents the Fretwell et al. (2013) surface elevation. We use Bamber et al. (2009) 500 m surface elevation contours and the grounding line from Bindschadler et al. (2011) (green). (b) Map of the continent with the location of the two stations indicated in red. Source: QuAntarctica.

operated year-round, including the winter period when PE is unmanned. The station and the set of instruments are controlled remotely via a satellite connection. Specifications of the instruments are given in Table 2 (see also Gorodetskaya et al. (2011, 2015)).

The Vaisala CL-31 ceilometer (firmware 1.72) was installed on the roof of the station in December 2009 and is operational at present. It emits laser pulses at central wavelength of $910 \pm 10$ nm at 298 K. The measurement resolution is set to 10 m and the reporting interval on 15 s. Several outages of the energy provision system limit the data mainly to Antarctic summer season (December to March is best represented). Only one year of continuous measurements was achieved (2015). We collocate information retrieved from the Micro Rain Radar to ceilometer blowing snow detection, to attest whether blowing snow happens during a precipitating event, using the return from the vertically profiling Doppler radar operating at a frequency of 24 GHz,



**Table 2.** Raw data and derived parameters of the instruments set up on the southern roof of Princess Elisabeth station,

| instrument | raw data | derived parameters |
| --- | --- | --- |
| Vaisala CL-31 ceilometer | attenuated backscatter vertical profiles | cloud base height and vertical extent, cloud phase, optical depth, blowing snow |
| Metek Micro-Rain radar 2 | spectral signal power per range | effective reflectivity, spectral width, mean Doppler velocity |
| Infrared radiation pyrometer Heitronics KT15.82 II | atmospheric brightness temperature | effective cloud base temperature |

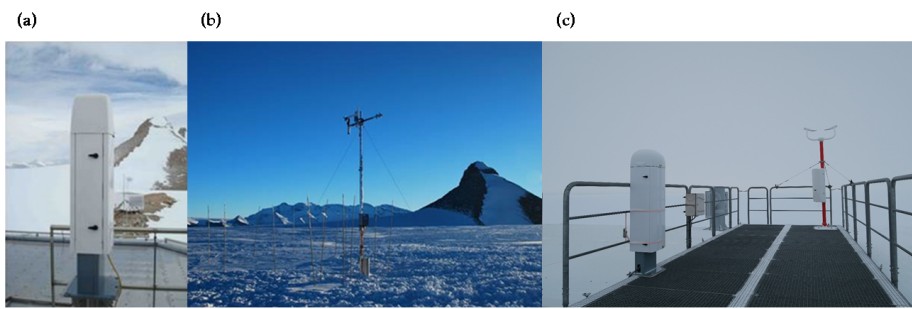

**Figure 3.** (a) the Vaisala CL-31 ceilometer on the roof of PE station, (b) the IMAU Automatic weather station at PE, (c) the Vaisala CL-51 ceilometer on the roof of Neumayer III station (credits:Hauke Schulz).

post-processed following Maahn and Kollias (2012) to link radar reflectivity to snowfall rates using the newly developed Ze-Sr relation for PE by Souverijns et al. (2017). We also use the atmospheric brightness temperature measurement from the infrared radiation pyrometer as cloud base temperature measurement.

An Automatic Weather Station has been set up 300 m from the station (71 °56' S; 23 °20' E) for recording meteorological pa-

5 rameters, broadband radiative fluxes and snow height changes (Gorodetskaya et al., 2013). It is designed to work continuously in remote locations, enabling studies of mass balance and radiative fluxes. The automatic weather station was established in February 2009, and replaced by a new station in December 2015, both designed by the Institute for Marine and Atmospheric Research, University of Utrecht (Utrecht, The Netherlands). The station provides hourly mean data of near ground and air temperature, a 1m profile of snow temperature (10 levels), air pressure, wind speed and direction, relative humidity and radiative

10 fluxes (downwards and upwards short- and long wave radiation), and records snow-height changes (for details on sensors, see Table S1 in Supplements). Post-processing of the data includes a treatment for relative humidities as described by Anderson



(1994) for humidity with respect to ice, and a correction for the relative humidities above 100 % following van den Broeke et al. (2004). The temperature gradient is computed as the difference between the 4m and surface temperatures over the distance between the sensors (m) (Gorodetskaya et al., 2013).

## 2.3 Neumayer III research station

Neumayer III research station is located on the Ekström ice shelf, in North East Weddell Sea (70 °40' S; 08 °16' W). Researchers are present year-round at the station and it is equipped with various instruments. Measurements include upper air soundings, ozone soundings, radiation measurements and weather observations. Weather observations are carried out since 1981 at Neumayer III, and the station is the weather forecasting centre for DML. The synoptic observations at Neumayer III include 2m and 10m air temperature, air pressure, wind vector at 2 and 10m height, 2m dew point temperature, presence - type and height of clouds, horizontal visibility, and past and present weather including snowdrift and whiteout (for a description of the sensors, see Table S2 in Supplements). The measurements are carried out routinely every 3 hours but visual observations are omitted at 03 and 06:00 UTC. In this paper we use the visual observations of blowing snow, classified into 9 categories (S8 code) according to the Word Meteorological Organization (WMO) coding system (see Table S3 in Supplements). The visual observations regarding blowing snow are performed as follows (detection procedure from Gert König-Langlo, personal communication, 2016): if the wind exceeds $5 \, \mathrm{m \cdot s^{-1}}$, the observer goes out about $100 \, \mathrm{m}$ wind ward from the research station and observes the snow surface. No target is used to detect blowing snow against, and during winter (no light at all), a small flashlight is used. The distinction between blowing and drifting snow is made according to the height of the blowing snow layer in relation to the eye level: drifting snow below the eye level, and blowing snow above. Further, if the blowing snow layer is not too dense, one can distinguish blowing snow with or without precipitaton by an additional observation from the roof of the station.

The set of instruments present at Neumayer III station includes a Vaisala ceilometer CL-51 (firmware 1.021), set up on the roof of the station and operating continuously since the 15th of January 2011. The ceilometer reports attenuated backscatters every $15 \, \mathrm{s}$ from 10 to $13\,500 \, \mathrm{m}$ height, with a vertical resolution of $10 \, \mathrm{m}$. The blowing snow record at Neumayer station is analyzed together with the atmospheric measurements available from the synoptic observations. The data (König-Langlo, 2010-2015) is freely available interactively from https://www.pangaea.de/.

## 3 Data treatment and blowing snow detection algorithm

### 3.1 Pre-processing

We average every 15s- $\beta_{att}$ profile over one hour using a running mean, to create mean attenuated backscatter profiles at every time step and get rid of turbulence and noise. Figure 4 shows the resulting $\beta_{att}$ at 09:30 UTC, based on the average of 240 profiles (120 preceding and 120 following 09:30 UTC). An additional reason for the integration of the signal over longer time periods, is that it improves the signal to noise ratio (SNR). No additional SNR correction is performed on the raw data, as we





found that a SNR higher than 0.3 would remove parts of the blowing snow signal (Gorodetskaya et al., 2015).

There are two sources of noise and artifacts affecting the ceilometer backscatter signal: the hardware of the Vaisala ceilometers, and the internal processing of the data (Kotthaus et al., 2016). Firstly, a heater is incorporated in the device to keep the instrument at a fixed temperature. This heater is placed close to the laser transmitter and the periodic turning on and off of the

heater introduces a small periodic variation in the stability of the emitted signal (and therefore of the detected signal). This effect is stronger in the first range bins, closest to the device. Secondly, the internal processing of the signal includes a built-in correction for the partial overlap of the laser in the first range bins. This overlap is due to the coaxial configuration of the laser: the same lens is used for the emitted and the received signals, made possible by the use of mirrors (Spirnhirne, 1993; Vande Hey, 2015). The total overlap is only reached at the 7th range bin (65 m) for the CL-31 (Kotthaus et al., 2016; Vande Hey,

2015). However, the partial overlap in the near-ground range bins does not imply that the minimum detection range is at 65 m only; in case the signal returned by the close range scatterers is large enough, it will be recorded even before the overlap onset (Vande Hey, 2015). Lastly, the CL-31 backscatter profile is constrained in the lowest bins by a built-in function to correct for unrealistically high values resulting from window obstruction. Yet, this correction likely introduces artifacts in the signal in the first range bins. As a result of the periodic switching on and off of the heater and the low overlap in the first range gate,

the reported value of the combined $\beta_{att}$ signal in the lowermost range bin is systematically and unrealistically higher than the signal in the next bins (Vaisala, personal communication, 2016). We therefore exclude the signal reported in the lowermost range bin in our analysis, and start investigating the profile from the second range bin 15 m above the CL-31 and CL-51 ceilometers onwards.

Moreover, artifacts have been observed in the ceilometer profiles at both stations (also visible in Fig. 5). There is a discontinuity

in all profiles between the 4th and the 5th range bins. This discontinuity is also visible in profiles where the instrument is completely covered, which are supposedly representing full attenuation, and thus recording the background noise produced by the hardware and electronics. Many authors have reported artifacts in the lowest range bins (below 70m height), that are usually excluded during processing for boundary layer investigation (Wiegner et al., 2014). This local minimum is also reported by Sokol (2014) at the 5th range bin during the whole duration of his campaign, as well as by Martucci et al. (2010) and Tsaknakis

et al. (2011). Kotthaus et al. (2016) states that these are likely due to the correction applied by Vaisala to prevent unrealistic values in the lower bins, related to the obstruction of the window and the internal noise. In the case of Vaisala instruments, the output is already corrected with a correction function, unknown to the user, and which cannot be modified (Wiegner et al., 2014). This has to be kept in mind when using the profile information to detect blowing snow.

### 3.2   The blowing snow detection algorithm

Studies investigating the boundary layer properties based on ceilometer attenuated backscatter make use of both properties of the signal: its shape and its intensity, to evaluate of the presence and extent of a particular layer. E.g. in order to determine the height of the mixing layer (Wiegner et al., 2014). For such analysis, five methods have been developed (Emeis et al., 2008), including a threshold method and a gradient method (Eresmaa et al., 2006). In the first case, the mixing height is attained when the intensity of the signal drops below a fixed threshold value (Münkel and Rasanen, 2004). The second method considers the

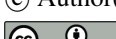



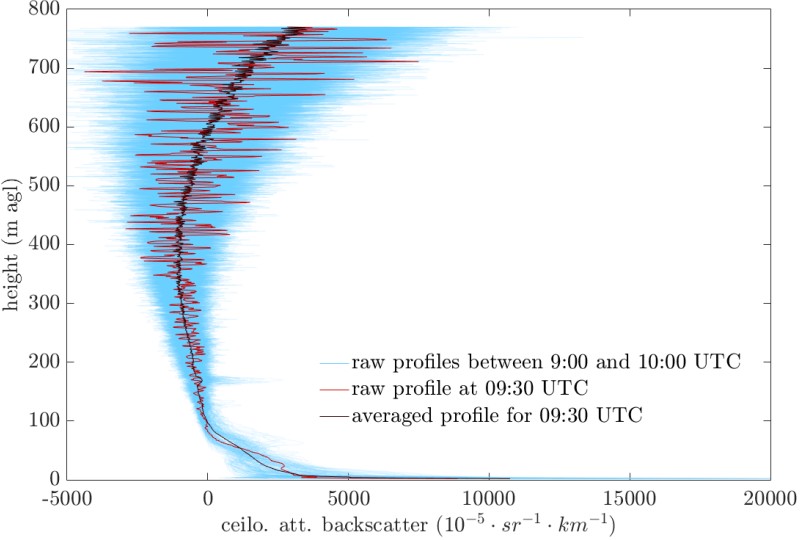

**Figure 4.** Hourly averaging of the attenuated backscatter profile of the CL-31 at PE. The attenuated backscatter profile at 09:30 UTC (red line) and resulting averaged profile (black) for the same timestep, based on the average of all the 240 profiles in blue.

minimum of the first or second derivative of the backscattering profile as top of the mixing layer (Sicard et al., 2004). To detect the occurrence of blowing snow, Palm et al. (2011) uses a combination of both types of methods on the CALIOP satellite backscatter. First, the intensity of the backscatter in the bin closest to the detected ground return exceeds a certain threshold. Second, the decrease of the profile of the signal with height indicates the presence of blowing snow: the concentration of

particles close to the ground is much higher than in the overlying layers (Takeuchi, 1980; Schmidt, 1982; Palm et al., 2011). This is associated with a sharp vertical gradient where the $\beta_{att}$ profile decreases strongly in the very first range bins. In addition, a wind speed threshold is applied (3 m·s$^{-1}$ at 10 m).

The approach used here is similar, but there is no wind speed criterion in our analysis. In addition, the ceilometer is ground-based, allowing the detection of blowing snow mixed with snowfall. To detect blowing snow, the intensity of the backscatter

signal at the lowest usable bin must exceed a certain threshold, and the intensity of the signal must decrease in the next range bins indicating a particles density greater in the lower levels than at the top of the layer. As previously highlighted, clean air molecules cannot be distinguished because the signal associated with it is smaller than the noise generated by the hardware (Wiegner et al., 2014; Kotthaus et al., 2016) and by the background light (Vande Hey, 2015), polluting the signal in the lowest bins. To distinguish the presence of scatterers (aerosols, blowing snow particles, cloud particles...) present in the atmosphere

from these artifacts, we need to investigate the signal intensity representative for clear sky conditions. I.e., the average $\beta_{att}$ of the second range bin received by the ceilometer during scatterer-free conditions. Clear sky days are selected using the daily quicklooks (Fig. 1) and are days where the quicklook background is uniform and without precipitation or clouds, and where the time series of the signal is stable around a low value (corresponding to hardware and background noises), to avoid low-level

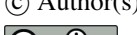


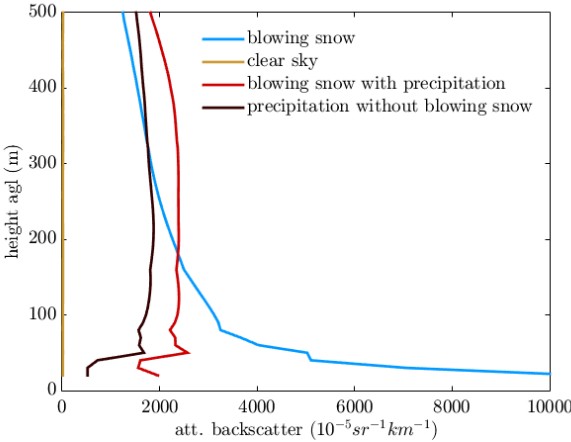

**Figure 5.** All types of profiles measured by the ceilometer at PE station on 24/04/2016. Typical blowing snow signal with no precipitation nor clouds in blue, blowing snow overlaid by precipitation in red. The black line represents precipitation with no blowing snow, and the yellow line shows the near-zero signal for clear sky conditions.

disrupting signal. Next, we select the clear-sky $\beta_{att}$ signal in the second range bin, and compute the 99th percentile as threshold value (for calculation, see section 3.3). As such, it is representative of the presence of scatterers exceeding the value for clear sky. Since the noise is instrument-dependent, individual pre-processing and thresholds have to be defined for each instrument the BSD algorithm is applied to.

If the value of the backscatter signal in the second range bin exceeds the clear-sky threshold, the BSD algorithm investigates the shape of the $\beta_{att}$ profile. A regular clear sky ceilometer profile (signal intensity versus height) does not show intense vertical variations (Fig. 5): in the infrared, the transmission term is close to one and decreases only slightly with height. This implies that any important variation in the $\beta_{att}$ signal can be attributed to the particles backscatter. The blowing snow and blowing snow with precipitation lines in Fig. 5 are typical blowing snow profiles: the blue line shows a sharp decrease until  100 m

height, above which the signal keeps decreasing steadily : this is the signature of clear sky blowing snow. The red profile, on the other hand, shows a re-increase in intensity a bit below 100 m height overlying the blowing snow signal: this indicates the presence of scatterers. If there is no blowing snow event, while precipitation is present, the profile does not decrease prior to the increase at higher levels (black line in Fig. 5). The discontinuity, as described in section 3.1., is also detectable in Fig. 1(b) and in all profiles in Fig. 5 between 40 and 50 m high. We therefore set as condition, that a blowing snow profile implies that

the mean of the overlying bins 3 to 7 (25 to 65 m) must be lower than the signal in the second range bin (15 m).

 Inherent to this profile-based method, the detection of blowing snow during precipitating events is limited to cases when the blowing snow signal is preserved close to the ground. In case of strong precipitation associated with storms, there is always blowing snow due to the high wind gusts displacing the fresh snow, and no distinction between precipitation and blowing snow is possible. The precipitation intensity might cover the blowing snow signal, even close to the ground. Then, the profile of

the backscatter intensity does not decrease with height, and the BSD algorithm does not detect blowing snow. Such events are



therefore not considered by the BSD algorithm.

In addition to the detection of blowing snow, the BSD algorithm quantifies the height of the layer. This is done as follows; if the profile decreases steadily (indication of absence of precipitation), the range gate at which the intensity of $\beta_{att}$ drops under the clear sky threshold value is the top of the layer. Anything above this height is considered clear sky. If there is precipitation

or a cloud during the blowing snow event, the shape of the backscatter profile does not decrease monotonously, but shows an increase in higher levels. In that case, the range gate at which the profile increases again is the top of the blowing snow layer, and the base of the cloud and/or precipitation.

### 3.3 Application of the blowing snow detection algorithm to different stations

The BSD algorithm is designed to detect blowing snow events reaching heights of minimum 15 m and is developed for the

Vaisala CL-31 located at PE station, for the period 2010-2016. It is applicable to other ceilometers: we applied the BSD algorithm to backscatter data from the Vaisala CL-51 ceilometer at Neumayer station, for the years 2011-2015. The time (15 s) and height resolution (10 m) is the same for both instruments. We can therefore apply the BSD algorithm in the same fashion to both datasets with the only difference being the attenuated backscatter threshold. We obtain a threshold of $21 \cdot 10^{-5} \cdot \mathrm{km}^{-1} \cdot \mathrm{sr}^{-1}$ for the CL-31 ceilometer at PE, based on 127 clear sky days out of a total of 1064 days. The threshold at Neumayer is of 32.5

$\cdot 10^{-5} \cdot \mathrm{km}^{-1} \cdot \mathrm{sr}^{-1}$, based on 125 clear sky days out of 1444 days.

## 4   Results

### 4.1   Frequency of blowing snow

In order to investigate the type of blowing snow detected by the BSD algorithm, we compare it to visual observations at

Neumayer. The WMO visual observations are categorized in six classes of blowing and/or drifting snow events, ranging in intensity and whether there is precipitation or not (Table S3 in Supplements). Before we start the comparison, it should be noted that visual observations are difficult to perform, and the error associated with it is not quantified. Therefore, in this part we refer to the number of measurements that match or mismatch between the BSD algorithm and visual observations rather than using the visual observations as "ground truth". The total number of measurements, N, is the total number of visual

observations performed during which the ceilometer is measuring, independently of whether there is blowing snow or not. The match ratio is the total agreement between visual and BSD algorithm detections over N; with $\mathrm{N}_{BSboth}$ when both the ceilometer and the observer detect blowing snow, and $\mathrm{N}_{BSnone}$ when neither the ceilometer nor the observer detects blowing snow. Mismatches occur when only one of the methods detects blowing snow, when the other does not : $\mathrm{N}_{BSceilo}$ if blowing snow is only reported by the BSD algorithm, and $\mathrm{N}_{BSvis}$ when only the visual observations record blowing snow :

$$match = \frac{N_{BSboth} + N_{BSnone}}{N}; \qquad\qquad mismatch = \frac{N_{BSceilo} + N_{BSvis}}{N} \qquad\qquad (2)$$

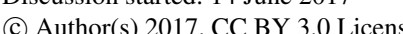



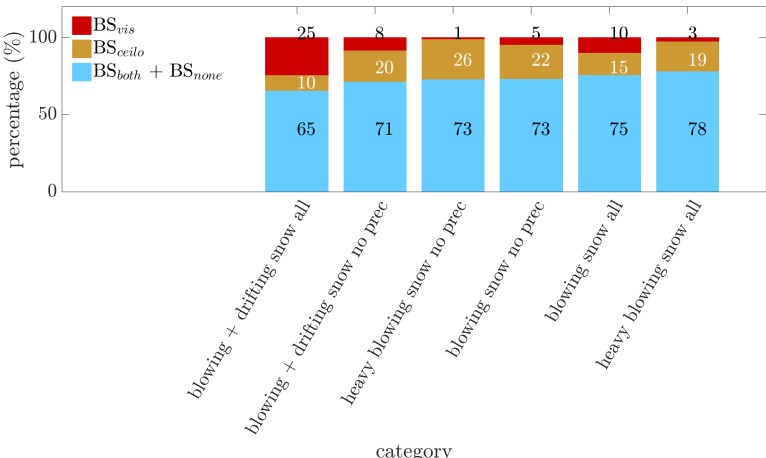

**Figure 6.** Ratio of matches (blue bar) compared to the ratio of mismatches (red and yellow bars) for each of the blowing snow categories. The matches encompass the $N_{BSboth}$ and $N_{Bsnone}$ and the mismatches are $N_{Bsceilo}$ in yellow and $N_{Bsvis}$ in red.

The results show a very good match in the blowing snow detection (Fig. 6) and the optimum, 78 %, is reached for events classified as heavy blowing snow with or without precipitation. The lowest match (65 %) is found when all blowing and drifting snow is taken into account : the number of visually detected events strongly increases since more categories are included, whereas the number of detections by the BSD algorithm is fixed. For this category covering everything, in 25 % of the cases the observer reports something that is not detected by the BSD algorithm (Table S4, in Supplements). This is related to the fact that the ceilometer points upwards, which prevents it from detecting shallow layers of drifting snow.

Over all visually detected events, the BSD algorithm detects 79 % of the heavy blowing snow events :

$$\frac{N_{BSboth}}{N_{BSboth} + N_{BSvis}} \tag{3}$$

In this case, we consider visual observations reported as 'heavy blowing snow' only. For 95 % of the $N_{BSceilo}$ events not reported as 'heavy blowing snow' by the observer, intensities of the backscatter signal are below $2000 \cdot 10^{-5} \cdot \mathrm{km}^{-1} \cdot \mathrm{sr}^{-1}$; it is therefore likely that those events are classified as 'slight' or 'moderate' by the visual observer instead of being considered heavy (visible in Fig.S3, in Supplements). For the $N_{BSvis}$, 54 % do not attain the threshold indicating the presence of scatterers and in 46 % of the cases the ceilometer attenuated backscatter profile does not decrease with height. Details on the division in the match and mismatch categories between $N_{BSboth}$, $N_{BSnone}$, $N_{BSceilo}$ and $N_{BSvis}$ are presented in Fig.S3 and Table S4 (Supplements).

The BSD algorithm output is binary at Neumayer: either there is blowing snow or there is no blowing snow, and no distinction can be made as whether there is precipitation or not, since precipitation measurements are not available at the station. The visual observer does however indicate whether there is precipitation or not. To investigate to which extend the BSD algorithm is limited by precipitation, we compare matches and mismatches for the heavy blowing snow category. The value fo $N_{BSboth}$





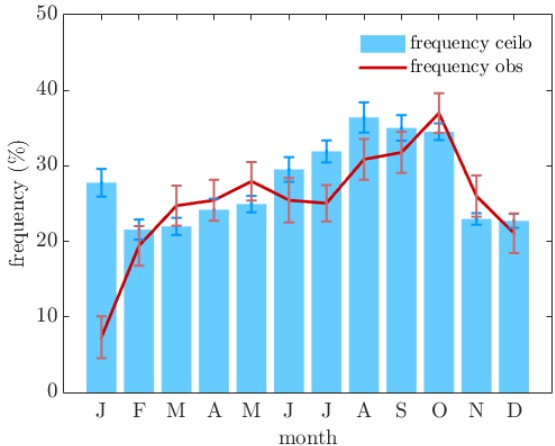

**Figure 7.** Annual cycle of blowing snow frequency at Neumayer III station, derived by the BSD algorithm (blue bars), and visual observations (red line) for the period 2010-2015. The error bars represents the inter annual variability.

triples when including events occurring during precipitation while $N_{BSceilo}$ decreases by nearly a third and $N_{BSvis}$ increases by one third. This indicates that the BSD algorithm is not impeded by the presence of precipitation.

Moreover, we gather that the ceilometer algorithm is not limited to heavy blowing snow, but that it also detects a number of visually detected events referenced under "moderate", or "slight blowing snow events", and even occasionally "drifting

snow". This is revealed by the fact that $N_{BSceilo}$ reduces as we consider less intense and shallower type of events (Table S4, in Supplements).

Blowing snow at Neumayer III occurs on average 28% of the time, as detected by the BSD algorithm (2011-2015). This is consistent with König-Langlo and Loose (2007), who report drifting and blowing snow frequency of 40%, and 20 % for blowing snow only. However, there is a strong inter annual variability in monthly blowing snow rates (Lenaerts et al., 2010).

The frequency is calculated here by reporting the sum of all hours during which blowing snow occurs (n = 2 714 164) over the total number of observation hours ( n= 9 742 717). The pattern visible in Fig. 7 is common for blowing snow over Antarctica: a seasonal cycle peaking during the Antarctic winter (March - November) and displaying lower values for the rest of the year (Mahesh et al., 2003; Lenaerts et al., 2010; Scarchili et al., 2010; Palm et al., 2011). The overall blowing snow frequency at PE equals 9 %, which is lower than at Neumayer, but is reasonable for the location of the station: PE is shielded from the katabatic

winds by the Utsteinen mountain range. The mean annual wind speed at the Princess Elisabeth station (5 m·s$^{-1}$) is lower compared to Neumayer (9 m·s$^{-1}$ , König-Langlo and Loose (2007)). The frequency retrieved here is coherent with Palm et al. (2011) and analogous to the situation of the Norwegian Troll research station (72 °00' 41" S - 2 °32' 06" E ) as detected by satellite.





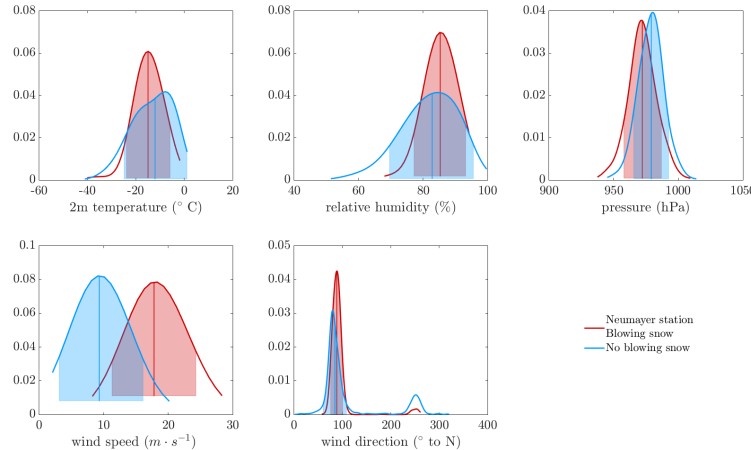

**Figure 8.** kernel probability density function of the atmospheric variables at Neumayer. The blue and red curves correspond to non-blowing snow and blowing snow conditions, respectively. The vertical line represents the median, and the shaded area the IQR of the density function, colors according to blowing or non-blowing snow. Bandwith = 4 for all variables.

## 4.2 Near-surface atmosphere changes during blowing snow

Analysis of meteorological conditions for blowing snow events compared to the rest of the time shows that the 2m wind speed, and mean relative humidity (RH), exhibit statistically significant differences (Figs 8 and 9; Tables S5 and S6, in Supplements). The 2m wind direction shows a preferential Easterly orientation at Neumayer and PE during blowing snow events, while non-blowing snow takes place under a wider spectrum of wind directions (Fig. 8 and 10). Part of the non-blowing snow measurements occur during katabatic conditions, when the wind blows from the interior towards the coast. Easterly winds during non-blowing snow conditions are probably related to the synoptic events during which no blowing snow occurs, or during which precipitation is too intense to conserve the blowing snow signal. Positive anomalies in wind speed, RH and incoming long wave radiation at the surface are associated to warm synoptic events, when air masses originating from the easterly winds bring moist air from the ocean precipitating inland. Such events are a common feature at Neumayer (König-Langlo and Loose, 2007) and occur 41-48 % of the time at PE (Gorodetskaya et al., 2013, 2014). Further, wind speed and RH are both conditions privileging blowing snow, but are also impacted by the blowing snow itself: wind speeds are high enough to be able to lift and bring the snow particles from the surface to drift and saltation. Then, the concentration of particles suspended in the atmosphere brings an extra friction, increasing the roughness length and reducing the wind speed (Bintanja and Reijmer, 2001; King and Turner, 1997). The increase in RH is both a result of moist air advection during synoptic events, and due to the sublimation of precipitation and blowing snow (Bintanja and Reijmer, 2001), a self-limiting process (Bintanja, 2001). This in turn lowers the air temperature close to the ground (King and Turner, 1997). At PE, the air temperature varies only slightly during blowing snow events, but the surface temperatures show a bigger increase as synoptic events are often



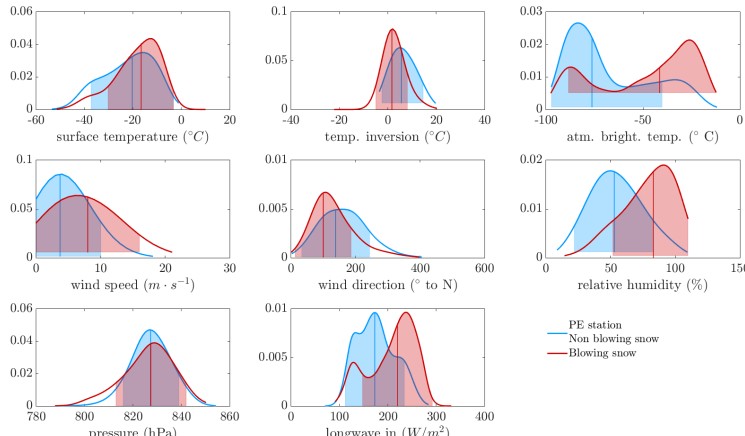

**Figure 9.** Kernel probability density function of the atmospheric variables at PE station. The blue and red curves correspond to non-blowing snow and blowing snow conditions, respectively. The vertical line represents the median, and the shaded area the IQR of the density function, colors according to blowing or non-blowing snow. Bandwiths: surface temperature: 5; temperature inversion: 4; atmospheric brightness temperature: 5; wind speed: 4; wind direction: 35; relative humidity: 10; pressure: 4; incoming long wave radiation: 10.

accompanied with clouds and high winds. Those also have an impact on the radiative budget. Finally, the air is less stratified as under katabatic conditions and the vertical temperature gradient is therefore lower as the air mixes during synoptic regimes and blowing snow events.

These variables are similar to those found by Gorodetskaya et al. (2013) categorizing the regimes at PE and enable to classify most of the blowing snow events with the warm synoptic regime bringing precipitation and storm, and the transition from this regime to the katabatic conditions. However, a few blowing snow events also occur in clear sky cold conditions, when the wind blows from the interior towards the coast, building up a stable boundary layer.

## 4.3 Blowing snow and meteorological regimes

In order to differentiate between dry blowing snow, and blowing snow associated wit precipitation, we analyze the influence of the wind, air and atmosphere brightness temperature, RH, temperature gradient, time since last precipitation and blowing snow layer height during blowing snow conditions by means of a principle component analysis. At both Neumayer and PE, the dominating parameters are the wind direction, followed by the time since last precipitation and the height of the layer as explanatory factors for the variability.

The wind direction is the dominating component, but does not explain the variations within blowing snow in itself. Rather, the wind direction is linked to the type of event and we can distinguish between clear sky blowing snow and events occurring together with storm and/or precipitation. A cluster analysis (for details, see Gorodetskaya et al. (2013)) is applied on blowing snow conditions at PE. Blowing snow there occurs mainly during synoptic, or transitional conditions (n = 461; 61 %), often



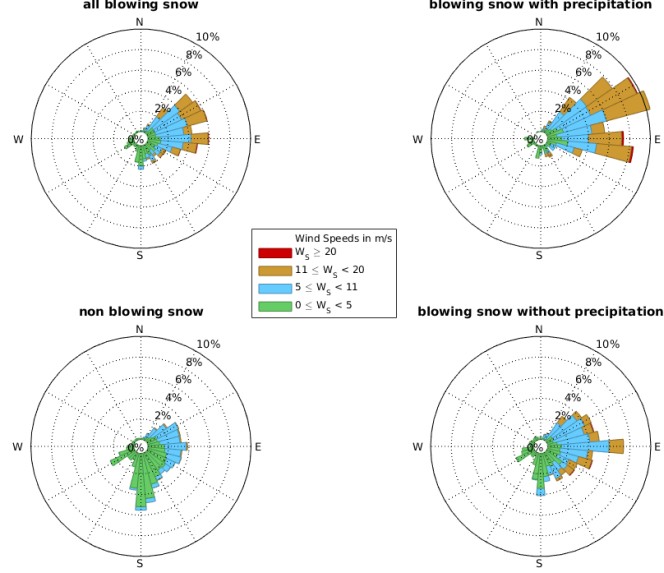

**Figure 10.** Wind roses presenting the wind direction for all blowing snow, blowing snow with or without precipitation , and non-blowing snow conditions at Princess Elisabeth station.

accompanied with precipitation. The attenuated backscatter signal is the highest during this type of events. The added value of the BSD algorithm over satellite detection is that those events are successfully detected by the algorithm, whereas satellite detection is limited to clear sky conditions, implying that a great part of the events during the synoptic regime would be missed, although they represent more than half of the events observed at PE. Less blowing snow is observed during the cold katabatic

regime (n = 165; 22 %), as can be seen on the wind rose (Fig. 10). The atmosphere is more stable, with a larger temperature inversion (in agreement with Gorodetskaya et al. (2015)) and the mean blowing snow layer thickness is lower. Blowing snow without precipitation, but with dominant easterly wind direction can be associated with transitional conditions (n = 126; 17 %), when the time lag since the last precipitation event is longer.

**4.3.1  Time since last precipitation and blowing snow occurrence**

The majority of blowing snow occurs during or within a day after a precipitation event (nearly 60 and over 80 % of the blowing snow occurrences, respectively). There is a clear drop for larger time lags (Fig.11(a)). This is, however, not so obvious anymore if we normalize the distribution of blowing snow events taking into account the total number of measurements within each time lag after precipitation (Fig.11(b)). A possible explanation is that the number of observations decrease with time,

and that blowing snow occurred during those observations. This can also be linked to the fact that the blowing snow particles





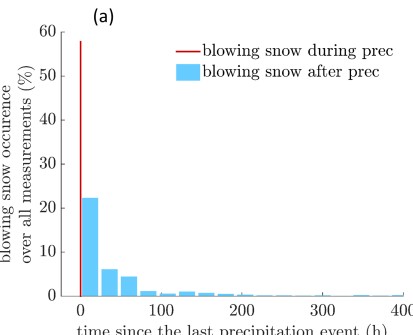
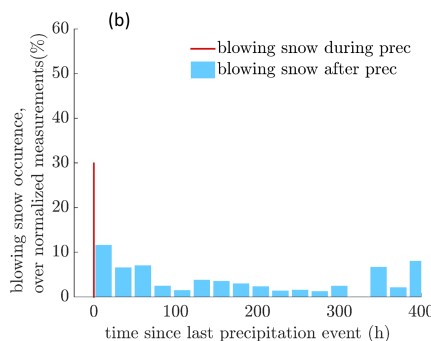

**Figure 11.** (a) Time between blowing snow and the last precipitation event at PE station. The red bar represents blowing snow occurring during precipitation, and the blue bars represent the fraction of blowing snow occurring each 24h time lag after a precipitation event. (b) Ratio of the number of blowing snow hours happening within the time lag over the total number of measurements for this time lag.

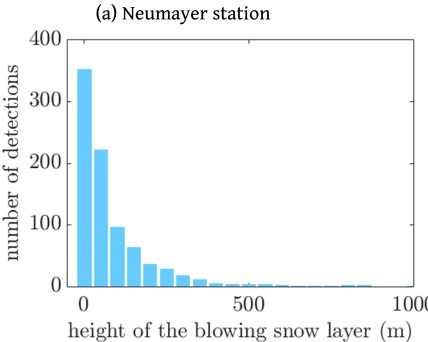
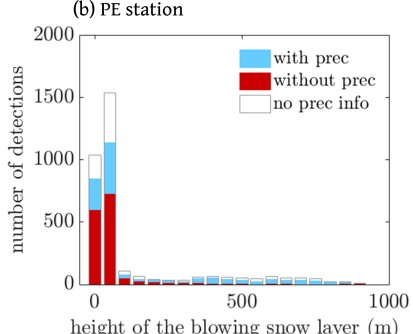

**Figure 12.** Distribution of the height of the blowing snow layer at (a) Neumayer station (b) at PE station, blowing snow accompanied with precipitation in blue, blowing snow without precipitation in red and the white bar represents periods with missing precipitation data.

detected by the BSD algorithm might originate from another location where there is precipitation, while no snowfall is detected by the precipitation radar at the station itself.

### 4.3.2  Depth of the blowing snow layer

The height of the blowing snow layer varies according to different parameters: wind speed, and the size and density of the snow particles. In addition, the presence of clouds and precipitation also influences the vertical extent of the blowing snow layer. Blowing snow layer depths at Neumayer III and PE show a predominance of shallow layers (over 65 and 75 % thicknesses below 100 m, respectively, Fig. 12). Blowing snow during precipitation at PE induces in general layers of higher vertical extend: mean layer height during precipitation reaches 234 m, while clear sky mean blowing snow layer depth is limited to 74 m. The values found for both stations are consistent to the mean blowing snow layer height detected by ground-based lidar at South Pole (Mahesh et al., 2003), although somewhat lower. The thickness of the blowing snow layer detected by the





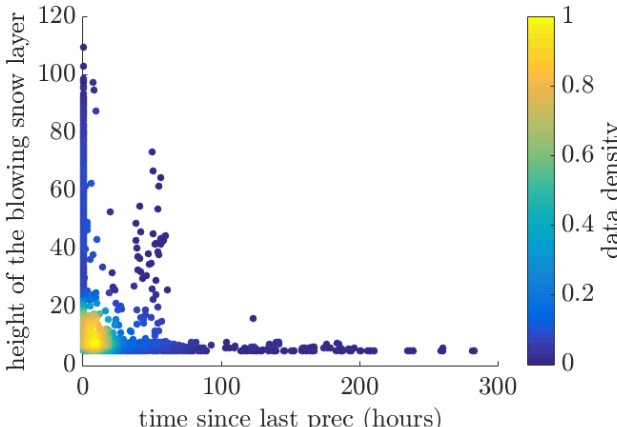

**Figure 13.** Scatter plot of the time since last precipitation event versus height of the blowing snow layer. Each point represents a blowing snow event. The colorbar represent the data density (number of observations divided by the entire sample size).

BSD algorithm is probably underestimated in case of heavy blowing snow events, due to total attenuation of the signal before reaching the top of the layer.

We further tested the hypothesis that the height of the blowing snow layer is related to wind speed. While there is no correlation, (also found by Mahesh et al. (2003)), the height of the blowing snow layer is related to the time since last precipitation (Fig.

13). The height of the blowing snow layer can reach up to 1000 m within 24 to 48 h after precipitation, and 95 % of the blowing snow layers thicker than 500 m occur shortly after the last precipitation event. Blowing snow events taking place much later after the precipitation event are limited to a vertical extend lower than 100 m thick.

## 5   Discussion

### 5.1   Applicability of the algorithm

The BSD algorithm developed for the Vaisala CL-31 ceilometer at PE was successfully applied to the Vaisala CL-51 ceilometer at Neumayer III station. Comparing the BSD algorithm detections to visual observations at Neumayer proved the applicability of the BSD algorithm to detect (heavy) blowing snow events, both under dry and precipitating conditions. The algorithm is able to detect blowing snow during most of the storms, which is an improvement compared to satellite detection. When we limit the analysis to (heavy) blowing snow, the algorithm detects 79 % of the events, indicated by the observer. On the other hand, there

are cases where the ceilometer does not detect events classified as heavy events by the observer. However, it has to be kept in mind that snowdrift observations are extremely challenging, with a potential large but unknown error on the observations. Furthermore, the hourly time filtering applied leads to commission errors (events detected that were not there) and commission errors (short-lived events are likely removed from the running mean). However, such events induce much smaller blowing



snow transport rates than strong events, and we suspect that omitting them will only reduce blowing snow transport rates by a small percentage. A limitation of the BSD algorithm is that both ceilometers are set up on the roof of the station, 17 m at Neumayer III and 12 m above the ground in the main wind direction at PE. In addition, 15 m have to be added to account for the discard of the first range bin. There, ceilometers will report the most significant blowing snow events (higher than 30 m)

and most drifting snow and shallow blowing snow events are not detected. Finally, due to the profile-shape based algorithm, the occurrence of blowing snow during severe storms or very heavy precipitation events can not be reported; then, the signal of blowing snow is mixed with that of precipitating snow, and the steady decrease of the profile until 45 m height is no longer valid.

The BSD algorithm can be applied to any ceilometer located in Antarctica, but we recommend to use a bin width of 10 m for

operating ceilometers to detect blowing snow, which is the case at PE and Neumayer III. Since the Vaisala CT25K at Halley station uses a 30m vertical resolution, it was not used in this study.

## 5.2 Wind speed versus snow availability

Gallée et al. (2001) stated that snow-pack properties mainly determine snow erosion: dendricity, density, sphericity and particles size regulate the availability of snow for transportation. These parameters change with metamorphism and impact the

friction velocity, and therefore the threshold friction velocity and minimum wind speed required for particles movement. Here, we do not apply any wind speed threshold to the detection of blowing snow, whereas many observations and modelling studies do so. Palm et al. (2011) uses a minimum wind speed criterion to detect blowing snow from satellite backscatter, potentially leaving out some events.

We find that the the presence of freshly fallen snow (availability and size/density of snow particles) has a great impact on

blowing snow occurrence and blowing snow layer height. As postulated by Mahesh et al. (2003), the end of a large snow storm with high wind speeds could still hold snow particles suspended in the air, even if the wind speed has already dropped to lower speeds than those required to dislodge the particles from the ground at the onset of the blowing snow event. Conversely, if no particles are available for the wind to pick up, blowing snow might not occur even though the wind speeds are high. Despite the fact that there is no mean to distinguish dry blowing snow from blowing snow occurring during precipitation from ground-

based instruments at Neumayer, the large majority of blowing snow events occur under synoptic disturbances (n = 867; 96 %) rather than katabatic conditions. These disturbances are also associated with higher wind speeds and are often accompanied with precipitation. In those cases, snow is available for transport. At PE, the explanation for the limited occurrence of blowing snow under katabatic conditions might lie in the fact that the station is shielded by the Sør Rondane mountains, but also due to the limited availability of fresh snow and the turbulence during those events, maintaining particles aloft. This might indi-

cate that the effect of katabatics on blowing snow occurrence has been overestimated, and the occurrence of synoptic events bringing fresh snow is a most determining factor for blowing snow.



## 6 Conclusions

Various observations, models and satellite studies have been performed to quantify and investigate blowing snow on the Antarctic continent. We present here our novel BSD algorithm, designed to retrieve blowing snow from ground-based remote-sensing ceilometers. The algorithm has proven to be reliable in detecting (heavy) blowing snow at Neumayer station in up to 79%

of the cases when compared to visual observations. The presence of precipitation does not substantially limit the retrieval by the ceilometer. This is an improvement to satellite detection, limited to clear sky conditions and therefore missing a great part of the blowing snow as more than half of the blowing snow happens during a storm. We further conclude that most of the blowing snow events happen during or shortly after precipitation, brought to the continent by the easterly winds associated to synoptic systems. The availability of fresh snow determines the onset of blowing snow, and the available fresh snow can be

lifted to higher heights than during katabatic conditions whose effect is likely to have been overestimated for lifting snow from the surface. This highlights again the limitation of wind speed thresholds, when applied to blowing snow retrieval methods, and the need to take into account the properties of the snow particles, including the availability of fresh snow, in order to accurately initiate blowing snow in models. Since ceilometers are low-cost robust instruments, and often deployed at stations for the purpose of aircraft operations, our newly developed algorithm opens opportunities for long-term monitoring networks

of consistent blowing snow observations. These can further be used to validate satellite retrieval and combined to produce blowing snow products over the ice sheets.

## 7 Code availability

The algorithm is freely available upon request to alexandra.gossart@kuleuven.be

## 8 Data and availability

Data from Neumayer station are freely available on the Pangaea portal and data from the instruments at Princess Elisabeth station are available upon request (www.aerocloud.be).

*Acknowledgements.* We are grateful to the Research Foundation Flanders (FWPO) and the Belgian Federal Science Policy (BELSPO) for the financial support of the AEROCLOUD project (BR/143/A2/AEROCLOUD). We thank the logistic team and the Royal Meteorological Institute for executing the yearly maintenance of our instruments at the Princess Elisabeth station. We further thank Wim Boot, Carleen

Reijmer, and Michiel van den Broeke (Institute for Marine and Atmospheric Research Utrecht) for the development of the Automatic Weather Station, technical support and raw data processing. We warmly thank World Radiation Monitoring Center for providing the Baseline Surface Network Radiation data set at Neumayer station, and Gert König-Langlo for the CL-51 ceilometer data and information about the visual observations. We further thank the Norwegian Polar Institute for the use of the free Quantarctica package, as well as Bindschadler et al. (2011); Bamber et al. (2009) for the datasets.



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
