# Peer review of "Blowing snow detection from ground-based ceilometers: application to East Antarctica"

_The Cryosphere, 2017_

## Referee Comment (RC1) · Anonymous Referee #1 · 29 Jun 2017

Multi-year observations from ground-based ceilometers are used to study blowing snow at two coastal locations in East Antarctica. The blowing snow products include notably monthly and annual frequencies of occurrence, blowing snow layer heights and time elapsed since last precipitation event. For one location the frequencies are evaluated against visual observations for a 6-year period (2010-2015) and show general agreement.

General comments: The paper is generally well written, although a more rigorous attention is required in some parts when describing and discussing the blowing snow processes. The results are interesting and original and can be of great potential but a substantial revision is needed before the paper becomes acceptable for publication. More specifically, I have some reservations on the profile classification procedure in its cur-

rent form. The distinction between precipitation and mixed blowing snow–precipitation events (Fig. 5) is not convincing. Information is lacking on how precipitation data are used to identify the occurrence of precipitation, as well as on the availability of data over the measurement period at PE. The (monthly and annual) frequency of occurrence is not studied at PE despite 7 years of measurements. Other potentially valuable information may be produced such as the inter-annual variability in blowing snow frequency (at both locations) or the relative proportion of mixed and pure blowing snow events (at least at PE). If you can use your profile classification to discriminate between blowing snow and mixed blowing snow events at Neumayer, this would be also of great interest. More generally, some parts need clarification and/or rerrangement, and the switching between different notions or locations make the manuscript sometimes difficult to read. Section 4.2 is not very useful. The conclusion, as well as the abstract, could contain more of the main (potential) results (annual and monthly frequencies, inter-annual variability, relative proportions of mixed blowing snow events, mean blowing snow layer heights). Indicate also in the abstract the respective locations (Neumayer/PE) and the time period to which your results correspond to. I recommend that all co-authors carry out thorough reading of the paper before resubmission.

Specific comments: 1. P2, L1-4: Despite the abundant literature on that topic, I recommend not to use wind speed ranges as a criterion to distinguish between drifting and blowing snow. As it is mentioned in the paper, the occurrence of drifting and blowing snow is strongly related to surface snow properties, which make the characterization of wind speed thresholds relative to the local climate conditions. For instance, low wind speeds can initiate erosion where loose snow is frequently brought by snowfall, while high wind speeds are needed to erode consolidated snow. The actual turbulent quantity involved in aerodynamic entrainment of surface snow particles is the friction velocity. Erosion starts when the actual friction velocity (depending on atmospheric flow conditions and surface aerodynamic properties) exceeds a threshold friction velocity (related to surface physical snow properties: density, cohesion, grain size, etc.). In the context of this paper, using a more general classification by mentioning just the

height at which windborne snow is observed is a more convenient way to describe the drifting (< 2m) and blowing snow (> 2m) processes. Besides, it is not correct to discriminate between suspension and blowing snow. Suspension is a transport mode and refers to diffusion of snow particles in the atmosphere picked up at the top of the saltation layer by turbulent eddies. For a given erosion event, the maximum elevation reached by suspended particles in define the height of the blowing sow layer, which is thus not necessarily confined to a few meters above the surface. Saltation is the other main transport mode, and describes ballistic trajectories and periodic rebounds of particles at the surface. Drifting and blowing snow thus must be seen as differently balanced situations between these two transport modes: drifting snow more generally refers to a situation where saltation is the dominant transport mode, while blowing snow stands for the opposite.

2. P2, L10: Similarly, the threshold speed of 11 m s-1 given by Kodama et al. (1985) is relative to the measurement period and location in Adelie Land and should not be presented as a general threshold above which the influence of snowdrift sublimation on SMB become significant.

3. P2, L11: This is not always the case. Change for "can be more effective".

4. P2, L17: "Affecting [. . .] the surface energy balance", not "affect [. . .] on surface . . .".

5. P2, L34: You should refer to Trouvilliez et al. (2014), who also report drifting snow statistics in East Antarctica from ground-based measurements with Flow-Capt instruments, instead of Trouvilliez et al. (2015) who present an evaluation of the Flow-Capt in the French Alps. The paper of Barral et al. has been published in 2014.

6. P5, L7 to P6, L2: These sentences belong to the methodology and should be moved in section 3.2.

7. P6, L3: How do you use this information in the study?

8. P7, L19: Distinguishing visually between pure blowing snow and mixed blowing

snow-precipitation events seems far too subjective to me, even if "the blowing snow layer is not too dense".

9. P8, L31: the first "of" has to be removed. Change "layer. E.g." for "layer, e.g."

10. P10, L5 and onwards: It is likely that I don't understand correctly the detection principle, but in its current form I have some reservations about your classification procedure, especially about the distinction between precipitation with and without blowing snow, and the omission of strong precipitation associated with heavy blowing snow. I tried to list them below. It is difficult to relate the profile features described in the text using heights and bin numbers to the plotted profiles in Fig. 5. You could, for instance, clearly indicate the discontinuity between the 4th and 5th bins, and specify to which bin the lowermost backscatter intensity value reported on the graph correspond to. This would facilitate the understanding of the description of the detection algorithm.

- The increase in the backscatter signal between the first and the second bins in the mixed blowing snow profile in Fig. 5 is of vey small intensity compare to the one characterizing pure blowing snow. Except for this aspect, this profile seems very similar to the pure precipitation profile. Moreover, I suppose that a mixed profile should include both the signature of precipitation and blowing snow (strong signal close to the surface). Are you sure that this absence of the blowing snow signature does not simply imply that there is no blowing snow?

- L14: "between 40 and 50 m": give the corresponding bin numbers.

- L17: I don't understand why during strong precipitation associated with storms, the precipitation intensity might cover the blowing snow signal close to the ground. I'm wondering even further if the opposite would be true. The strong backscatter signal close to the surface in the typical blowing snow profile illustrates the influence of high particle density layers. This would be particularly amplified when abundant snowfalls provide a large supply of fresh snow that can be easily eroded by strong winds. By discarding these cases, you might omit an important part of the mixed blowing snow

events, which can further affect all your statistics. This could be a major issue since you say latter in the paper that most of the blowing snow events occur simultaneously with precipitation. If the situation with strong precipitation and blowing snow is a clear limitation of your approach, you have to quantify it, especially since the occurrence of overcast conditions is also a limitation to satellite retrieval. You should give the relative proportion of each profile category (blowing snow, precipitation + blowing snow, precipitation, clear sky and omissions).

11. P11, Section 4.1: There is a temporal discordance between visual observations (performed 6 times a day) and ceilometer measurements (hourly means). Have you re-sampled the ceilometer dataset to match the frequency of visual observations, or do you compare the ceilometer hourly output corresponding to the time at which the visual observations were performed? Are the visual observations continuous over the measurement period (2010-2015)?

12. P12, Figure 6: Indicate N for each category.

13. P12, L16: Don't you think you could use the profile classification developed at PE (in terms of vertical variation in backscatter intensity) to discriminate the occurrence of precipitation at Neumayer?

14. P12, L19: An "r" is missing in the penultimate word.

15. P13, Fig. 7: How can you explain the apparently systematic discordance between visual observations and the detection algorithm in January?

16. P13, L7: Please indicate over which period of time the frequency is computed.

17. P13, Second paragraph: this paragraph is hard to follow and needs rearrangement:

- L9-11: You switch between annual and monthly time scales, and frequency and blowing snow rates. Move the sentence in which you describe the calculation of the frequencies at the beginning of the paragraph. Indicate the time period over which König-Langlo and Goose (2007) computed their frequencies. Remove "blowing snow

rates" and stay focus on frequencies to compare apples and apples. Indicate also the measurement period for the frequency computed at PE (and for this you also need to discuss the representativeness of the winter data due to power supply issues).

- L13: See also Trouvilliez et al. (2014) and Amory et al. (2017) for similar statistics from ground-based measurements.

- L14: "Reasonable" is not rigorous. Please replace.

- L16: In the previous sentence you give the frequency for two locations (Neumayer: 28% and PE: 9%): which one do you compare with Palm's results? "Coherent" and "analogous" give no quantitative information, and are somewhat confusing when used together. Give directly the values from Palm et al. (2011) (and indicate the measurement period) and, then, discuss the particular geographical settings of PE to explain the contrast in wind speed and, ultimately, in blowing snow frequencies, with the other results/locations mentioned in the text. If the frequencies compare reasonably well with satellite measurements, does this mean that the hindering effect of clouds is not so influent? Again this appears contradictory with the apparently frequent occurrence of precipitation and overcast conditions during blowing snow events.

18. P14, Fig. 8 (legend): Non blowing snow (not "no")

19. P14, section 4.2: This section could have been more organized. You alternate between katabatic and synoptic conditions, blowing snow and non-blowing snow conditions, PE and Neumayer, and results and theory. Some sentences are ambiguous, others contain syntax errors, irrelevant or incomplete information, and some conclusions seem a bit early. I think you could remove this section entirely without disturbing your global analysis. Moreover, this would avoid redundant information with section 4.3, in which you actually refer to the work of Gorodetskaya et al. (2013) to define the two meteorological regimes. Find more detailed comments below:

- L5: "Fig. 8 and 10": an "s" is lacking

- L5-7: You only use a wind direction criterion to distinguish katabatic from synoptic conditions. What about a combined influence of katabatic and synoptic conditions? Is the deflection due to the Coriolis force also an influent factor accounting for the easterly component of the surface flow?

- L8-10: This sentence is ambiguous. Please rephrase.

- L11-13: Harsh construction. The colon (":") is misused. "wind speeds are high enough to be able to. . .and saltation" is clumsy: I guess "wind speeds are high enough to initiate snowdrift" is analogous but more concise.

- L12: The increase in RH is (partly) caused by blowing snow, not a cause of, so it doesn't "privilege" blowing snow.

- L13-15: Mentioning the self-limiting process of blowing snow sublimation and the increase in roughness due to windborne snow particles is not relevant since i) they are not a result here and ii) they don't explain any described feature.

- L15-16: This sentence needs rephrasing: "The increase in RH is both a result [. . .] and sublimation (not "due to") of precipitating and blowing snow particles."

- P15, L1: "Those also have an impact on the radiative budget": This is elusive. Illustrate and discussed further or remove.

- P15, L2: Turbulent mixing generally occurs during strong winds, whatever their origin (synoptic or katabatic). How do you distinguish between synoptic and katabatic conditions?

- P15, L4: "These variables": You mean "trends" (?)

20. P16, Fig. 10 (caption): Indicate the relative proportion of each category.

21. P16, L4: "as", not "although".

22. P16, L13: Remove "anymore".

23. P17, section 4.3.2: It is not clear how the depth of the blowing snow layer is determined.

24. P18, Fig. 11 (ordinate axis): Indicate the units.

25. P18, L10: If your algorithm is applied "successfully", then you consider the visual observations as ground truth. Compare favorably with or something like that, would be more appropriate. Idem for "proved the applicability".

26. P19, L15: Metamorphism does not impact the friction velocity, only the threshold friction velocity (see comment #1).

27. P19, L17: Can you give more examples of such (many) studies?

28. P19, L19: a "the" is redundant. The properties listed in brackets are not complementary information of "freshly fallen snow". Please rephrase.

29. P19, L29: Which role do you give to the turbulence during katabatic conditions in limiting the occurrence of blowing snow at PE?

30. P19, L29-31: Katabatic winds or conditions, not "katabatics". Please clarify where and how the effect of katabatic winds on the occurrence of blowing snow has been over-estimated? Do you actually mean that katabatic winds are not the main driven force behind blowing snow at PE, as usually considered? If so, you should limit this conclusion to the particular geographical settings of PE, which are likely non-representative of the general conditions in coastal East Antarctica.

31. P20, L7: Specify that this conclusion is only valid for PE.

32. P20, L9: "mainly determines".

33. P20, L10-11: In which context this conclusion has been drawn?

34. P20, L12: "The availability": you mean erodibility (availability of fresh snow is not a snow property)?

35. P20, L15: Use "evaluate" rather than "validate".

References:

Amory, C., Gallée, H., Naaim-Bouvet, F., Favier, V., Vignon, E., Picard, G., Trouvilliez, A., Piard, L., Genthon, C., and Bellot, H.Âă: Seasonal variations in drag coefficient over a sastrugi-covered snowfield in coastal East Antarctica. Boundary-Layer Meteorol., 164, 107–133, 2017.

Barral, H., Genthon, C., Trouvilliez, A., Brun, C., Amory, C.: Blowing snow in coastal Adelie Land, Antarctica: three atmospheric-moisture issues, The Cryosphere, 8, 5, 1905–1919, 2014.

Kodama, Y., Wendler, G., and Gowink,J.: The effect of blowing snow on katabatic winds in Antarctica, Ann.Glaciol., 6, 59–62, 1985.

Trouvilliez, A., Naaim-Bouvet, F., Genthon, C., Piard, L., Favier, V., Bellot, H., Agosta, C., Palerme, C., Amory, C., and Gallée, H.: A novel experimental study of aeolian snow transport in Adélie Land (Antarctica), Cold Reg. Sci. Technol., 108, 125–138, 2014.

Trouvilliez, A., Naaim-Bouvet, F., Bellot, H., Genthon, C., and Gallée, H.: Evaluation of FlowCapt acoustic sensor for snowdrift measurements, J. Atmos. Ocean. Technol., 32, 1630–1641, 2015.

---

## Referee Comment (RC2) · Anonymous Referee #2 · 17 Jul 2017

Report on the paper "**Blowing snow detection from ground-based ceilometers: application to East Antactica**", by alexandra Gossart et al.,

This paper present a new application of the ceilometer, namely the detection of blowing snow events in Antarctica. As blowing snow measurements in Antarctica are quite scarse and because blowing snow is potentially an important contributor to the antarctic surface mass balance, this study must be considered.

The paper is divided into a technical part and an analysis of the results which is based on a comparison of ceilometer output with visual observations at Neumayer station, where a ceilometer was also deployed.

Results are up to now restricted to a count of the number of events and focuss on strong events, referred to as blowing snow in the paper, since the ceilometer analyses vertical profiles and since it is localised on the roof of both buildings of Neumayer and Princess Elizabeth (PE) station.

The calibration of the ceilometer remains an issue and I wonder if it will be possible. This is important since the use of a ceilometer would start to be relevant for blowing snow studies if a quantitative retreiving of blowing snow characteristics (height of the blowing snow layer, amount of transported snow) may be done. Is it the intention of the authors to perform such a calibration in the future? This point must be considered in the discussion and the conclusion in order to advice the reader about the potentialities and weaknesses of the study.

Nevertheless the paper represents a sufficient amount of work to be published. To my opinion the technical decription should be improved for a TC reader, especially a modeller. In fact there is too much or not enough. An alternate possibility should be to shorten the technical description and to do it in an other more specialized journal.

I had difficulties with a double meaning of some sentences (see specific comments below).

Specific comments:

p. 2, line 5. : the word "suspension" is defined here but is no used in the rest of the paper (see e.g., line 18 p.7), so that its intoduction here is not clear.

p.2, line 24, note that the precipitation process is also poorly constrained in Antarctica so that the authors have to face to one equation on SMB with at least two unknowns: precipitation and snow erosion by the wind.

p.3, section 2. What are the altitude of both stations PE and Neumayer. Is their climate (e.g., SMB, summer temperature, …) different? This will help the reader when considering the development of the BSD by using observations at Neumayer and using it for another location.

p.3, section 2.1. An introductory sentence stating that the ceilometer was not initially set up for measuring blown snow events would clarify the section. More generally the description here should contain more information related to a possible use of the measurements for a determination of blown snow characteristics.

p.4, line 1. : what is the raw resolution in time of the ceilometer?

p.4, line 2. : "spatial resolution": do you mean "vertical"?

p.4, line 20. : please indicate for each instrument which measurement you intend to use in the paper, especially concerning the infrared pyrometer (see also p.6, line 2 where it is not said there for which purpose the cloud base height deduced from the brighness temperature is used). As for the next comment it is preferable to describe the use of an instrument in a single paragraph.

p. 5, lines 4-6. The ceilometer is described twice. Please rearrange the text.

p.5, lines 8-9. Please clarify the description of the MRR.

p.8, line 4. Please indicate the reason of the turning on/off of the heater.

p.14, line 13. What about the role of sastrugi in the evolution of blowing snow intensity?

p.14 – 15, fig. 8 and 9. How do you quantify from a statisticall point of view the differences between blowing snow and non blowing snow wind speed and relative humidity? What is your interpretation of the differences for the other variables?

p.16, line 2. What is the avantage of satellite detection?

p.16, line 14. Clarify "observations"

p.18, lines 17 – 18: "commission errors": please clarify.

p.19, line 2. Is it possible to improve the set-up of the ceilometers on the field, and how?

p.20, line 3. : … designed to retreive blowing snow events but no drifting snow from ground-based ...

---

## Referee Comment (RC3) · Anonymous Referee #3 · 27 Jul 2017

**Summary**

This manuscript presents a new algorithm to identify blowing snow in clear-sky as well as during precipitation (not possible for satellite-based estimation) in Antarctica using vertical profiles of backscatter coefficient values from ceilometers. The depth of the blowing snow layer can also be estimated. The algorithm is first described and evaluated by comparison with collocated human observations collected at the Neumayer III station over several years. Given the satisfactory performance, the method is then applied to another data set collected at the Princess Elizabeth station. Statistics about the occurrence and depth of the blowing snow layers are derived, and the links with local meteorological conditions as well as weather regimes are investigated. These analyses reveal that the time to the last precipitation event is a key factor for the occurrence and depth of blowing snow layers.

**Recommendation**

This manuscript presents an original algorithm that will be useful to increase the pool of data to better understand the occurrence and dynamics of blowing snow. The study of the links between blowing snow occurrence and depth with meteorological conditions is interesting and provide new perspectives on the main factors controlling those features of blowing snow. The data and methods are well described and seem solid, although the evaluation could be based on more robust criteria and the details of the algorithm should be better illustrated. I hence recommend to send the manuscript back to the authors for moderate revisions. I have some comments and suggestions, listed below.

**General comments**

1. The evaluation of the performance of the detection algorithm is based on the comparison with human observations at Neumayer III. The different classes (occurrence or not) may be unbalanced (much more cases without blowing snow than with, as suggested on l.14, p.13) requires more robust statistics than the one used. There is a lot of literature about what criteria can be employed for such confusion matrices. See for instance *Allouche et al.* (2006). I suggest the authors to sue such commonly used statistics (e.g. Cohen's kappa, true skill statistics) for the evaluation of the algorithm. The estimated depth is not really evaluated, what would be needed to do so?

2. The statistics derived from the outcome of the blowing snow detection algorithm
are informative and relevant, but they could be more complete, by including data and analysis about the inter- and intra-event variability of the blowing snow occurrence and depth.

**Specific comments**

1. P.7, l.28: the choice of smoothing the signal over 1 h should be better justified (why 1 h and not 30 min or 2 h?). The typical variability of the BS layer features should be commented (if there is a lot of dynamics within 1 h, one may loose relevant information by smoothing over 1 h).

2. P.8, l.1: "SNR higher than 0.3": I guess it is expressed in dB. If so, it should be clearly mentioned.

3. P.10, Fig.5: I probably missed something, but I do not understand why the backscatter signal from BS+precip is so much smaller (below 100 m alt) than the one from BS only (red vs blue). I would expect the two signals to sum up somehow... Or is the concentration in BS particles much smaller when there is precip? If so, what could be the explanations?

4. P.10, l.19: related question: it is written "The precipitation intensity might cover the blowing snow signal", which I find confusing with the curves in Fig.5 (for the lower altitudes). To be clarified...

5. P.11, l.6-7: about the estimation of the top of the BS layer: it would help the reader to indicate in Fig.5 where is this limit. And how reliable would be the outcome in case of virgas?

6. P.12, l.7-14: better metrics could be computed to evaluate the performance of the algorithm, see General comments above.
7. P.14, l.3: Which statistical tests have been used to check if there are "statistically significant differences"?

8. P.15, l.16: a minimum of description should be provided about the clustering method employed, so the reader does not have to check the reference to know what type of clustering method has been used for instance...

9. P.16, Fig.10: the font size of the text in the figure should be increased.

10. P.16, l.14: I do not understand why the number of observations would decrease... The ceilometer is collecting data every 15s, no? The explanation should be clarified.

11. P.18, Fig.13: are these distributions for the two stations (Neumayer and PE) or only one location?

12. P.18, l.17: "commission errors" is repeated twice.

13. P.19, l.9: I guess the same algorithm could be applied to lidar systems, no?

**References**

Allouche, O., A. Tsoar, and R. Kadmon (2006), Assessing the accuracy of species distribution models: prevalence, kappa and the true skill statistic (tss), *J. Anim. Ecol.*, *43*(6), 1223–1232.

---

## Author Comment (AC1) · 13 Sep 2017

**Response to Reviewer 1 Comments:**
**Blowing snow detection from ground-based ceilometers: application to East Antarctica**

Alexandra Gossart[1], Niels Souverijns[1], Irina V. Gorodetskaya[2,1], Stef Lhermitte[3,1], Jan T.M. Lenaerts[4,1,5], Jan H. Schween[6], Alexander Mangold[7], Quentin Laffineur[7], and Nicole P.M. van Lipzig[1]

[1]Department of Earth and Environmental Sciences, KU Leuven, Leuven, Belgium
[2]Centre for Environmental and Marine Sciences, Department of Physics, University of Aveiro, Aveiro, Portugal
[3]Department of Geosciences and Remote Sensing, Delft University of Technology, Delft, the Netherlands
[4]Institute for Marine and Atmospheric research Utrecht, Utrecht University, Utrecht, The Netherlands
[5]Departement of Atmospheric and Oceanic Sciences, University of Colorado, Boulder CO, USA
[6]Institute of Geophysics and Meteorology, Koeln University, Koeln, Germany
[7]Royal Meteorological Institute of Belgium, Brussels, Belgium

For clarifying our answers to the referees' comments, the following scheme is used: comments of the referees are denoted in **bold**, our answers are denoted in black and quotes from the revised text are in *italic*. Please note that reference to figures in the answer refer to the original manuscript, or to the improved figure displayed in the Response document. Figures referenced in the italic text are relative to the new manuscript.

**General comments: The paper is generally well written, although a more rigorous attention is required in some parts when describing and discussing the blowing snow processes. The results are interesting and original and can be of great potential but a substantial revision is needed before the paper becomes acceptable for publication. More specifically, I have some reservations on the profile classification procedure in its current form. The distinction between precipitation**

10 **and mixed blowing snow – precipitation events (Fig. 5) is not convincing. Information is lacking on how precipitation data are used to identify the occurrence of precipitation, as well as on the availability of data over the measurement period at PE. The (monthly and annual) frequency of occurrence is not studied at PE despite 7 years of measurements. Other potentially valuable information may be produced such as the inter-annual variability in blowing snow frequency (at both locations) or the relative proportion of mixed and pure blowing snow events (at least at PE). If you can use your**

15 **profile classification to discriminate between blowing snow and mixed blowing snow events at Neumayer, this would be also of great interest. More generally, some parts need clarification and/or rearrangement, and the switching between different notions or locations make the manuscript sometimes difficult to read. Section 4.2 is not very useful. The conclusion, as well as the abstract, could contain more of the main (potential) results (annual and monthly frequencies, inter-annual variability, relative proportions of mixed blowing snow events, mean blowing snow layer heights). Indicate**

20 **also in the abstract the respective locations (Neumayer/PE) and the time period to which your results correspond to. I**

**recommend that all co-authors carry out thorough reading of the paper before resubmission.**

Thank you for your thorough and advised comments. First of all, the methodology has been revised and the profiles classification is now used to detect the presence of clouds and/or precipitations from the ceilometer attenuated backscatter signal shape only. This enables to conduct the analysis of dry blowing snow versus blowing snow mixed with snowfall at both sta-
5   tions. Additionally, cases where blowing snow is mixed with heavy snowfall are also identified and occur 67 % of the cases at Neumayer III and 43 % of the cases at PE station, while 25 - 27 % of the events take place under cloudy or precipitating events. Cloudless blowing snow is rare (8%) at Neumayer III station, and reaches 30 % at PE. Figure 5 has also been adapted by choosing more adequate examples. Concerning the results and the detailed comments, please find individual answers below. Regarding the availability of the data, a graph is added to the Supplements (Fig. S2). Frequencies at PE are also present in
10  the Supplements (Fig. S1). However, since only one year of full measurements is available, the Antarctic winter cannot be studied. For instance, the high frequency in July 2015 and lower frequencies for the May, June, and August months are not robust enough. Regarding the inter-annual blowing snow frequency, it is not displayed in the Annual cycle at PE, but is present in Fig. 7 in the original manuscript as the error bar and is in the range of $\pm 5\%$. Section 4.2 has been removed. Abstract and conclusion contain now more of the main results.

**1 Question 1: P2, L1-4: Despite the abundant literature on that topic, I recommend not to use wind speed ranges as a criterion to distinguish between drifting and blowing snow. As it is mentioned in the paper, the occurrence of drifting and blowing snow is strongly related to surface snow properties, which make the characterization of wind speed thresholds relative to the local climate conditions. For instance, low wind speeds can initiate erosion where loose snow is frequently brought by snowfall, while high wind speeds are needed to erode consolidated snow. The actual turbulent quantity involved in aerodynamic entrainment of surface snow particles is the friction velocity. Erosion starts when the actual friction velocity (depending on atmospheric flow conditions and surface aerodynamic properties) exceeds a threshold friction velocity (related to surface physical snow properties: density, cohesion, grain size, etc.). In the context of this paper, using a more general classification by mentioning just the height at which windborne snow is observed is a more convenient way to describe the drifting (< 2m) and blowing snow (> 2m) processes. Besides, it is not correct to discriminate between suspension and blowing snow. Suspension is a transport mode and refers to diffusion of snow particles in the atmosphere picked up at the top of the saltation layer by turbulent eddies. For a given erosion event, the maximum elevation reached by suspended particles in define the height of the blowing sow layer, which is thus not necessarily confined to a few meters above the surface. Saltation is the other main transport mode, and describes ballistic trajectories and periodic rebounds of particles at the surface. Drifting and blowing snow thus must be seen as differently balanced situations between these two transport modes: drifting snow more generally refers to a situation where saltation is the dominant transport mode, while blowing snow stands for the opposite**

The paragraph in the paper has been changed accordingly, not referring to wind speeds thresholds, and with a more refined reference to saltation and suspension modes in blowing and drifting snow.

*Snow particles can be dislodged from the snow surface, picked up by the wind and lifted from the ground into the near-surface atmospheric layer. This phenomenon occurs approximatively on 70% of the Antarctic continent during winter (Palm et al., 2011). Generally, drifting snow events are shallower than blowing snow events. Drifting snow typically stays below 2 m height whereas blowing snow can reach heights of several hundreds of meters. The transport involves a mix of suspension and saltation transport modes (Leonard et al., 2011), with a dominance of saltating particles (Bagnold, 1974) in the case of drifting snow, and suspended particles in blowing snow layers (Mellor, 1965).*

**2 Question 2: P2, L10: Similarly, the threshold speed of 11 m s-1 given by Kodama et al. (1985) is relative to the measurement period and location in Adelie Land and should not be presented as a general threshold above which the influence of snowdrift sublimation on SMB become significant.**

The threshold wind speed has been removed.

*However, blowing snow is crucial for the regional SMB (Lenaerts and van den Broeke, 2012; Déry and Yau, 2002; Gallée et al. , 2001; Groot Zwaaftink et al., 2013) through the displacement and relocation of the snow particles (Déry and Tremblay, 2004). In addition, sublimation contributes substantially to SMB (Takahashi et al., 1992; Thiery et al., 2012; Dai and Huang, 2014; Kodama et al., 1985). This process can even be more effective to remove mass than surface sublimation (van den Broeke et al., 2004).*

**3   Question 3: P2, L11: This is not always the case. Change for "can be more effective".**

The text has been adapted accordingly.

*See question 2 above.*

**4   Question 4: P2, L17: "Affecting [. . .] the surface energy balance", not "affect [. . .] on surface . . .".**

The text has been adapted accordingly.

*Blowing snow also plays a role in determining snow surface characteristics (Déry and Yau, 2002), affecting the surface energy balance (Lesins et al., 2009; Mahesh et al., 2003; Yamanouchi and Kawaguchi, 1985).*

**5   Question 5: P2, L34: You should refer to Trouvilliez et al. (2014), who also report drifting snow statistics in East Antarctica from ground-based measurements with Flow-Capt instruments, instead of Trouvilliez et al. (2015) who present an evaluation of the Flow-Capt in the French Alps. The paper of Barral et al. has been published in 2014.**

The references have been changed accordingly

*A number of measurement campaigns have been organized in various regions of the AIS, using different types of devices: nets, mechanical traps and rocket traps, photoelectric and single-beam photoelectric sensors. Various studies have also worked with Flow-Capts or piezoelectric devices (Leonard et al., 2011; Amory et al., 2015; Trouvilliez et. al., 2014; Barral et al., 2014).*

**6   Question 6: P5, L7 to P6, L2: These sentences belong to the methodology and should be moved in section 3.2**

These sentences have been moved to section 3.2

Section 2.2
*The Vaisala CL-31 ceilometer (firmware 1.72) was installed on the roof of the station in December 2009 and is operational at present. It emits laser pulses at a central wavelength of $910 \pm 10$ nm at 298 K. The measurement resolution is set to 10 m*

*and the reporting interval on 15 s. Several outages of the energy provision system limit the data mainly to Antarctic summer season (December to March is best represented). Only one year of continuous measurements was achieved (2015). The Metek vertically-profiling precipitation radar, set up since 2010, enables to retrieve snowfall rates, using the return from the vertically*

5 *profiling Doppler radar operating at a frequency of 24 GHz. The raw Doppler spectra is post-processed following Maahn and Kollias (2012), to calculate radar reflectivity profiles which are then linked to snowfall rates using the newly developed Ze-Sr relation for PE by Souverijns et al. (2017) and has a sensitivity up to -14 and -8 dBz (Souverijns et al., 2017). A full description of micro-rain radars can be found in Klugmann et al. (1996) and the radar set up at Princess Elisabeth is described in Gorodetskaya et al. (2015).*

Section 3.2

*The information retrieved from the Micro Rain Radar (hourly precipitation rates) is collocated to ceilometer blowing snow detection, to determine the time (in hours) since the last precipitation event.*

**7 Question 7: P6, L3: How do you use this information in the study?**

15 The cloud base temperature is used as an atmospheric variable. In the case of blowing snow, the measured cloud temperature is actually the blowing snow layer temperature. It was used in the cluster analysis, and in the PCA. However, it was not a determining variable. This information is left out in the new version of the paper.

**8 Question 8: P7, L19: Distinguishing visually between pure blowing snow and mixed blowing snow-precipitation events seems far too subjective to me, even if "the blowing snow layer is not too dense".**

20 Yes, the visual detection of pure blowing snow versus mixed events is subjective. This method is applied by the visual observer at Neumayer station, following the procedure described by Gert König-Langlo (personal communication, 2016). This further reinforces our position of not treating the visual observations as "ground truth".

**9 Question 9: P8, L31: the first "of" has to be removed. Change "layer. E.g." for "layer, e.g."**

The text has been changed accordingly

25

*Studies investigating the boundary-layer properties based on ceilometer attenuated backscatter make use of both properties of the signal (shape and intensity), to evaluate the presence and extent of a particular layer, e.g. in order to determine the height of the mixing layer (Wiegner et al., 2014).*

**10**   **Question 10: P10, L5 and onwards: It is likely that I don't understand correctly the detection principle, but in its current form I have some reservations about your classification procedure, especially about the distinction between precipitation with and without blowing snow, and the omission of strong precipitation associated with heavy blowing snow. I tried to list them below. It is difficult to relate the profile features described in the text using heights and bin numbers to the plotted profiles in Fig. 5. You could, for instance, clearly indicate the discontinuity between the 4th and 5th bins, and specify to which bin the lowermost backscatter intensity value reported on the graph correspond to. This would facilitate the understanding of the description of the detection algorithm.**

- **The increase in the backscatter signal between the first and the second bins in the mixed blowing snow profile in Fig. 5 is of very small intensity compare to the one characterizing pure blowing snow. Except for this aspect, this profile seems very similar to the pure precipitation profile. Moreover, I suppose that a mixed profile should include both the signature of precipitation and blowing snow (strong signal close to the surface). Are you sure that this absence of the blowing snow signature does not simply imply that there is no blowing snow?**

- **L14: "between 40 and 50 m": give the corresponding bin numbers.**

- **L17: I don't understand why during strong precipitation associated with storms, the precipitation intensity might cover the blowing snow signal close to the ground. I'm wondering even further if the opposite would be true. The strong backscatter signal close to the surface in the typical blowing snow profile illustrates the influence of high particle density layers. This would be particularly amplified when abundant snowfalls provide a large supply of fresh snow that can be easily eroded by strong winds. By discarding these cases, you might omit an important part of the mixed blowing snow events, which can further affect all your statistics. This could be a major issue since you say later in the paper that most of the blowing snow events occur simultaneously with precipitation. If the situation with strong precipitation and blowing snow is a clear limitation of your approach, you have to quantify it, especially since the occurrence of overcast conditions is also a limitation to satellite retrieval. You should give the relative proportion of each profile category (blowing snow, precipitation + blowing snow, precipitation, clear sky and omissions).**

Figure 5 (Fig.6 in the new manuscript, Fig. 1 below) has been adapted to show both bin number and height (m agl). The discontinuity is clearly indicated in grey.

- Figure 5 is indeed not clear: it was based on only one day (24.04.2016) during which blowing snow was accompanied by clouds/precipitation at the end of the blowing snow event, hence, the lower intensities and the resemblance to the pure precipitation profile. Another day was therefore selected for the new version of the manuscript (10.02.2014), to illustrate the pure precipitation and the mixed event. In this new figure (Fig.1 below, Fig.6 in the new manuscript), the intensity of the profile in the lowermost bins is clearly indicative of blowing snow (red line), and the increase around the 15th bin indicates the presence of clouds/precipitations (arrow). In the case of precipitation/cloud without blowing snow, this low-level decrease is absent, and the increase around the 17th bin reflects the presence of a cloud/precipitation.

[Figure]

**Figure 1.** Different types of profiles relevant for blowing snow measured by the ceilometer at PE station: blue line - typical blowing snow signal with no precipitation nor clouds (24-04-2016); red line - blowing snow overlaid by precipitation (10-02-2016); black line - precipitation with no blowing snow (10-02-2014); yellow line - near-zero signal for clear sky conditions (24-04-2016). The height above ground is indicated on the right axis and the corresponding bin number on the left axis. All profiles exclude the lowermost bin, and start at the second bin (15 m agl.). The grey lines represent the discontinuity between bins 4 and 5 (35-45 m). The arrows indicate the presence of precipitation.

– The bin numbers have been added in Fig. 5 and in the text (see Fig.1 below, Fig.6 in the new manuscript).

– Given the specific conditions during heavy precipitation events, we treat these events differently in the improved manuscript. We know that most of the time, blowing snow happens together with storms and intense precipitation (the snowflakes rebound on the ground and are displaced by strong winds). Hence, in some cases the signal intensity is not decreasing with height, and the profile criterion could not be met. Therefore, we decided to create a new category "heavy mixed events" for the situation in which the signal in the second bin exceeds $1000 \cdot 10^{-5} \cdot \mathrm{km}^{-1} \cdot \mathrm{sr}^{-1}$ (threshold adapted from (Gorodetskaya et al., 2015)). Of those heavy mixed events, 45 % do not show an increase of the signal in the overlying bins at Neumayer III station, and would have therefore been discarded by the algorithm.

10 To conclude, a new method for precipitation/cloud detection based on the ceilometer profile only has been developed ( see Fig.2): the algorithm searches upwards of the 7th bin (maximum limit for the profile criterion of the BSD algorithm) for a second increase in signal that is (1) above $100 \cdot 10^{-5} \cdot \mathrm{km}^{-1} \cdot \mathrm{sr}^{-1}$, which is the threshold for clouds detection (Van Tricht et al. (2014)), and (2) thicker than 9 bins (85m) (Van Tricht et al., 2014). This enables to detect overcast conditions, in the

[Figure]

**Figure 2.** Chart of the method used to detect blowing snow from the attenuated backscatter signal of the ceilometer.

presence of blowing snow or not.

Regarding the intense mixed events, once the backscatter in the second range bin exceeds the blowing snow threshold, the algorithm evaluates if the threshold for intense mixed events is reached (above $1000 \cdot 10^{-5} \cdot \mathrm{km}^{-1} \cdot \mathrm{sr}^{-1}$, adapted from Gorodetskaya et al. (2015)). If it is the case, blowing snow is assumed present and the profile is not investigated.

*The algorithm therefore investigates the shape of the profile in order to detect blowing snow. A condition is set, that a blowing snow profile implies that the mean of the overlying bins 3 to 7 (25 to 65 m) must be lower than the signal in the second range bin (15 m). In this way, the discontinuity, as described in section 3.1. (visible in Figures 1 and 5 between 35 and 45 m in the original manuscript, Fig.1 and 6 in the new manuscript), is not affecting our retrievals. In order to detect blowing snow occurring during clouds or precipitation, the profile shape is analyzed to identify a second increases in the signal intensity above the 7th bin (65 m height). A clear differentiation between clouds or precipitation cannot be made on the basis of the ceilometer alone, but the presence of clouds and/or precipitation can be identified. This analysis is carried out for both blowing snow and non blowing snow measurements. [...]*

*Inherent to this profile-based method, the detection of blowing snow during precipitating events is limited to cases when the blowing snow signal is preserved close to the ground. In case of precipitation associated with storms, there is always blowing snow due to the high wind displacing the snow, and no distinction between precipitation and blowing snow is possible, as the ceilometer signal is entirely attenuated near the surface (Gorodetskaya et al., 2015), it is not possible to get signal in the overlying bins, and the profile of the backscatter intensity might not decrease upwards. Such intense precipitating events mixed with snowfall are identified as having a second bin signal higher than $1000 \cdot 10^{-5} \cdot \mathrm{km}^{-1} \cdot \mathrm{sr}^{-1}$ (threshold adapted from Gorodetskaya et al. (2015)). In those cases, the events are classified as a heavy mixed blowing snow event, and the profile analysis is eluded by the algorithm.*

**11 Question 11: P11, Section 4.1: There is a temporal discordance between visual observations (performed 6 times a day) and ceilometer measurements (hourly means). Have you re-sampled the ceilometer dataset to match the frequency of visual observations, or do you compare the ceilometer hourly output corresponding to the time at which the visual observations were performed? Are the visual observations continuous over the measurement period (2010-2015)?**

We have re-sampled ceilometer to hourly output, and selected the re-sampled data corresponding to the time at which visual observations are carried (1) if there are more than 140 measurements (35 mins) with a NaN value, the measurements within the hour are discarded. Else, if there is more than 20 mins of blowing snow detections, blowing snow is assumed for that measured hour (only to get rid of really short lived events). Then, we compare this with the visual observation.

Yes, the visual observation are continuous over the measurement period, but omit observations at 03 and 06:00 UTC.

*In order to investigate the type of blowing snow detected by the BSD algorithm, we compare it to visual observations at Neumayer, carried out routinely at 09-12-15-18-21 and 24:00. All ceilometer measurements are considered over one hour, corresponding to the time at which visual observations are carried out. We identify a blowing snow event when blowing snow is present in at least 80 profiles (20 mins). The WMO visual observations are categorized in six classes of blowing and/or drifting snow events, ranging in intensity and whether there is precipitation or not (Table S3 in Supplements).*

and section 2.3

*The measurements are carried out daily every 3 hours but visual observations are omitted at 03 and 06:00 UTC.*

**12 Question 12: P12, Figure 6: Indicate N for each category.**

The figure has been removed from the new version of the manuscript. Table 1 below (Table 3 in the new manuscript) lists the number of detections (N) for each category. The total number of event for each category is also displayed in the wind rose figures (Figs 3 and 4 below, Figs. 9 and 10 in the new manuscript).

**13 Question 13: P12, L16: Don't you think you could use the profile classification developed at PE (in terms of vertical variation in backscatter intensity) to discriminate the occurrence of precipitation at Neumayer?**

We have conducted this analysis (see Question 10), and similar trends as observed at PE station regarding blowing snow associated with precipitating events and synoptic disturbances.

*Further, we investigate the specific meteorological conditions (near-surface temperature inversion, relative humidity, surface temperature, wind speed and direction, in- and outgoing longwave fluxes, and the time since the last precipitation event)*

**Table 1.** Detection numbers and scores of the different categories of observations. The first 4 columns give N BS$_{both}$- stands for blowing snow detected by both the algorithm and the visual observations, N BS$_{none}$ - when both methods agree that there is no blowing snow, N BS$_{ceilo}$ and N BS$_{vis}$ - represent detections by the algorithm and the observer only, respectively (the corresponding percentages are presented in table S4, in the supplement). The four last columns give the scores. B stands for blowing and D for drifting snow. The total number of measurements is 10584.

| | N BS$_{both}$ | N BS$_{none}$ | N BS$_{ceilo}$ | N BS$_{vis}$ | accuracy | sensitivity | specificity | TSS |
|---|---|---|---|---|---|---|---|---|
| B and D snow, with or without prec | 2404 | 5170 | 972 | 2308 | 0.70 | 0.51 | 0.84 | 0.35 |
| B and D snow, without prec | 992 | 6578 | 2373 | 897 | 0.70 | 0.52 | 0.73 | 0.26 |
| heavy B snow, without prec | 378 | 7406 | 2998 | 72 | 0.72 | 0.84 | 0.71 | 0.55 |
| all B snow, without prec | 822 | 6993 | 2554 | 485 | 0.72 | 0.63 | 0.73 | 0.36 |
| all B snow, with or without prec | 1856 | 6665 | 1520 | 813 | 0.78 | 0.69 | 0.81 | 0.51 |
| heavy blowing snow, with or without prec | 1114 | 7249 | 2262 | 229 | 0.77 | 0.83 | 0.76 | 0.59 |

*during blowing snow events.*

*For all three categories of blowing snow events, the 2m wind direction shows a preferential easterly/north-easterly orientation at both Neumayer and PE, while the absence of blowing snow is characterized by a wider spectrum of wind directions*

5 *(Figs.3 and 4 below, Figs. 9 and 10 in the new manuscript). Positive anomalies in wind speed and RH occur during blowing snow events. Cyclonic events are a common feature at Neumayer (König-Langlo and Loose, 2007), bringing easterly winds during which most of the drifting and blowing snow occur. Also at PE, most of the blowing snow events (N = 1643, 92 %) are associated with the warm synopic and transitional regimes, when moist air is brought from the ocean, that precipitate inland (Gorodetskaya et al., 2013). Thiery et al. (2012) also showed that at PE drifting snow sublimation occurs mostly during*

10 *transitional regimes. These regimes occur 41-48 % of the time (Gorodetskaya et al., 2013, 2014). Very few blowing snow events occur in cloudless cold conditions (cold katabatic regime), when the northerly winds blows from the interior towards the coast (N = 139; 8%).*

*Intense mixed events ( Fig.1 above, Fig. 6 in the new manuscript) occur together with north-easterly strong winds : 87°to N, 10 m·s$^{-1}$ at PE and 65°to N, 13 m·s$^{-1}$ at Neumayer III , warmer surface temperatures and higher relative humidity. These*

15 *are the signature of storms associated with synoptic events, during which the turbulent mixing reduces the vertical temperature gradient (Gorodetskaya et al., 2013). The majority (60 %) of the blowing snow events occur during storms or overcast conditions (with cloud and/or precipitation). These mixed events have generally a short time lag since the last precipitation event and reach high atmospheric levels. Dry blowing snow has a mean wind direction of 120°to N at PE and 77°at Neumayer III, lower wind speeds (6-7m·s$^{-1}$) and a greater temperature inversion. The mean time lag since the last precipitation event*

20 *at PE (23 hours) indicates that these events most likely occur after a storm, and that cloudless blowing snow (8 %) is mostly associated to katabatic winds.*

[Figure]

**Figure 3.** Wind rose at PE station, N = number of events

[Figure]

**Figure 4.** Wind rose at Neumayer station, N = number of events

**14    Question 14 : P12, L19: An "r" is missing in the penultimate word.**

The "r" has been added.

**15    Question 15: P13, Fig. 7: How can you explain the apparently systematic discordance between visual observations and the detection algorithm in January?**

Indeed, January fall completely outside the variability of the other months in the visual observations. We suspect that there is some issue with these data in January, as no visual observations are reported in January 2011, 2013, and only a few are available during January 2014 and 2015. Other months, such as February 2011-2013 and 2015, as well as November and December 2014 and 2015 have also a restricted number of visual reports. We suspect that the observers might have been away on the field or not available for reporting during those periods. However, the ceilometer was operating continuously during these months. In addition, by sub-sampling the ceilometer blowing snow detections to the corresponding visual observation hours, the frequencies retrieved are biased (if a storm occurred between midnight and 09:00 UTC, it is not reported, and therefore excluded from the frequencies calculation). The frequency distribution presented here (Fig.5 below, Fig. 7 in the new manuscript) is therefore calculated on ceilometer measurements only, which are continuous over time, and are not compared to visual observations. The total frequency is of 36 %, and the reason this frequency is higher than in the previous manuscript, is that we now include heavy mixed events.

**16    Question 16 : P13, L7: Please indicate over which period of time the frequency is computed**

The period (2011-2015) was indicated. The sentence has been adapted to make it clearer.

*The frequency is calculated here by reporting the sum of all hours during which blowing snow occurs (n = 2 714 164) over the total number of observation hours ( n= 9 742 717). Blowing snow at Neumayer III occurs on average 28% of the time for the 2011-2015 period, as detected by the BSD algorithm. [...] The overall blowing snow frequency is computed at PE for the 2010-2017 period. However, the limited availability of Antarctic winter data (due to power failures at the station) might lead to an underestimation of the blowing snow frequency. Total blowing snow frequency reaches 13 % at PE station, which is lower than at Neumayer [...]*

**17    Question 17: P13, Second paragraph: this paragraph is hard to follow and needs rearrangement:**

  – **L9-11: You switch between annual and monthly time scales, and frequency and blowing snow rates. Move the sentence in which you describe the calculation of the frequencies at the beginning of the paragraph. Indicate the time period over which König-Langlo and Goose (2007) computed their frequencies. Remove "blowing snow**

[Figure]

**Figure 5.** Yearly cycle of blowing snow at Neumayer III station (2011-2015). The error bars represent the interannual variations.

**rates" and stay focus on frequencies to compare apples and apples. Indicate also the measurement period for the frequency computed at PE (and for this you also need to discuss the representativeness of the winter data due to power supply issues).**

5     – the frequencies paragraph has been changed accordingly.

    – **L13: See also Trouvilliez et al. (2014) and Amory et al. (2017) for similar statistics from ground-based measurements.**

    – the references have been added and the text was modified.

    – **L14: "Reasonable" is not rigorous. Please replace.**

10     – 'reasonable' has been rephrased

    – **L16: In the previous sentence you give the frequency for two locations (Neumayer: 28% and PE: 9%): which one do you compare with Palm's results? "Coherent" and "analogous" give no quantitative information, and are somewhat confusing when used together. Give directly the values from Palm et al. (2011) (and indicate the measurement period) and, then, discuss the particular geographical settings of PE to explain the contrast in wind speed and, ultimately, in blowing snow frequencies, with the other results/locations mentioned in the text. If the**

**frequencies compare reasonably well with satellite measurements, does this mean that the hindering effect of clouds is not so influent? Again this appears contradictory with the apparently frequent occurrence of precipitation and overcast conditions during blowing snow events.**

5    – The map present in (Palm et al., 2011) gives a range rather than a precise number. In the case of PE station, for instance, blowing snow frequency is 0-10 % while the BSD algorithm reaches 13 % of blowing snow (not 9 % since we include the heavy blowing snow events, the frequency increased). In this case, the BSD frequency is higher than the detection rate by the satellite method. This can be related to the number of blowing snow events occurring together with clouds/precipitation, missed by the satellite, and to the different spatial and temporal dimensions of the different

10      methods. In addition, the geographical settings of PE station are discussed.

*The frequency is calculated here by reporting the sum of all hours during which blowing snow occurs at Neumayer based on the BSD algorithm over the total number of observation hours. Blowing snow at Neumayer III occurs on average 36% of the time for the 2011-2015 period. This is consistent with König-Langlo and Loose (2007), who report 20 % of drifting and 40 % drifting and blowing snow for the 1981 - 2006 period. However, there is an inter-annual variability that reaches $\pm$ 5*

15 *% , also observed by Lenaerts et al. (2010). The pattern visible in Fig.5 above (Fig. 7 in the new manuscript) is common for blowing snow over Antarctica: a seasonal cycle peaking during the Antarctic winter (March - November) and displaying lower values for the rest of the year (Mahesh et al., 2003; Lenaerts et al., 2010; Scarchili et al., 2010; Palm et al., 2011; Amory et al., 2017). The overall blowing snow frequency is computed at PE for the 2010-2017 period and reaches 13%. This lower blowing snow frequency at PE can be explained by the location of the station: the station is shielded from the katabatic winds*

20 *by the Utsteinen mountain range, making it a quieter zone between the flows diverged to the sides of the station (Parish and Bromwich, 2007), while Neumayer III station is located on the ice shelf and experiences higher wind speeds [...] and is more exposed to storms. In addition, the limited availability of Antarctic winter data (due to power failures at the station) leads to an underestimation of the blowing snow frequency as mostly extended summer period was used, and only one winter is taken into account.*

25 *The frequencies measured by the BSD algorithm are larger than those retrieved by satellite method: Palm et al. (2011) gives a range of 0-10 % blowing snow for both locations. This can be related to the number of blowing snow events occurring together with clouds/precipitation, missed by the satellite, and to the different spatial and temporal dimensions of the different methods. Of all blowing snow detected events, 67 % is mixed with intense events at Neumayer III, and 43 % at PE station. Cloudless blowing snow is very rare at Neumayer III station (8 % of the events), while it reaches 30 % at PE station.*

30 **18    Question 18: P14, Fig. 8 (legend): Non blowing snow (not "no")**

The figure is not displayed in the new version of the manuscript.

**19    Question 19: P14, section 4.2: This section could have been more organized. You alternate between katabatic and synoptic conditions, blowing snow and non-blowing snow conditions, PE and Neumayer, and results and theory. Some sentences are ambiguous, others contain syntax errors, irrelevant or incomplete information, and some conclusions seem a bit early. I think you could remove this section entirely without disturbing your global analysis. Moreover, this would avoid redundant information with section 4.3, in which you actually refer to the work of Gorodetskaya et al. (2013) to define the two meteorological regimes. Find more detailed comments below:**

This section has been removed, only parts are kept in section 4.3. Separate answers are given for the remarks still present in the new version of the paper:

– **L5: "Fig. 8 and 10": an "s" is lacking**

– L5 : The figures are not displayed anymore

– **L5-7: You only use a wind direction criterion to distinguish katabatic from synoptic conditions. What about a combined influence of katabatic and synoptic conditions? Is the deflection due to the Coriolis force also an influent factor accounting for the easterly component of the surface flow?**

– L 5-7: There are three regimes: warm synoptic, cold katabatic, and transitional, when the situation evolves from synoptic to katabatic or the other way around as was defined by Gorodetskaya et al. (2013). While the wind direction was the dominant parameter in the PCA analysis, the parameters used to distinguish between these regimes are the wind direction, together with the temperature inversion and cloudiness, as well as the wind speed and relative humidity. Regarding the deflation to the East, ongoing analysis (Souverijns et al, in prep) showed that among the low pressure systems that are circling eastward around Antarctica over the Southern Ocean - mostly those centered to the north and to the northwest from PE determine the synoptic conditions at the PE station. As winds turn clockwise around the cyclone, air from oceanic areas is drawn towards the station. These oceanic air masses have the potential to take up a lot of moisture, and precipitate at the coastal areas of Dronning Maud Land, as winds are forced to rise against the Antarctic plateau. In those cases, winds at PE originate from the north east (when the cyclone is located to the northwest) or from the more inland areas at the east (when the low pressure system is located north of the station).

– **L8-10: This sentence is ambiguous. Please rephrase.**

– This sentence has been removed.

– **L11-13: Harsh construction. The colon (":") is misused. "wind speeds are high enough to be able to. . .and saltation" is clumsy: I guess "wind speeds are high enough to initiate snowdrift" is analogous but more concise.**

– The sentence has been rephrased accordingly.

– **L12: The increase in RH is (partly) caused by blowing snow, not a cause of, so it doesn't "privilege" blowing snow.**

- The sentence has been removed.

- **L13-15: Mentioning the self-limiting process of blowing snow sublimation and the increase in roughness due to windborne snow particles is not relevant since i) they are not a result here and ii) they don't explain any described feature.**

- The sentence has been removed.

- **L15-16: This sentence needs rephrasing: "The increase in RH is both a result [. . .] and sublimation (not "due to") of precipitating and blowing snow particles."**

- The sentence has been removed.

- **P15, L1: "Those also have an impact on the radiative budget": This is elusive. Illustrate and discussed further or remove.**

- The sentence has been removed.

- **P15, L2: Turbulent mixing generally occurs during strong winds, whatever their origin (synoptic or katabatic). How do you distinguish between synoptic and katabatic conditions?**

- PE station is shielded by the Utsteinen mountain range, therefore katabatic winds have the lowest wind speeds (see Fig.3 above and Fig. 10 in the new manuscript), compared to synoptic or transitional regimes.

- **P15, L4: "These variables": You mean "trends" (?)**

- Yes, the sentence has been adapted accordingly

*The near surface atmosphere changes, associated with blowing snow events, are investigated for both stations, and detailed means and standard deviation are displayed in Table S6 and S7, in supplements. We investigate how blowing snow hourly means relate to weather regimes, derived from the hierarchical cluster analysis applied in Gorodetskaya et al. (2013), which defines the weather regimes at PE station: "cold katabatic", "warm synoptic", and "transitional synoptic". The cold katabatic regime is characterized by slower wind speeds and lower humidity, reduced incoming long wave radiation, a slight surface pressure increase, and a substantial temperature inversion. Warm synoptic conditions involve higher wind speeds and specific humidity, strongly positive anomalies of incoming long wave radiation. The surface pressure is slightly lower, and the temperature inversion is strongly reduced than during average conditions. Finally, average wind speeds, humidity and incoming long wave radiation, as well as slightly lower surface pressure are observed during the transitional regime, when the situation evolves from synoptic to katabatic or the other way around (Gorodetskaya et al., 2013). Further, we investigate the specific meteorological conditions (near-surface temperature inversion, relative humidity, surface temperature, wind speed and direction, in- and outgoing longwave fluxes, and the time since the last precipitation event) during blowing snow events.*

*For all three categories of blowing snow events, the 2m wind direction shows a preferential easterly/north-easterly orientation*

*at both Neumayer and PE, while non-blowing snow takes place under a wider spectrum of wind directions (Figs. 9 and 10). Positive anomalies in wind speed and RH occur during blowing snow events. Cyclonic events are a common feature at Neumayer (König-Langlo and Loose, 2007), bringing easterly winds during which most of the drifting and blowing snow occur.*

5   *Also at PE, most of the blowing snow events (N = 1643, 92 %) are associated with the warm synoptic and transitional regimes, when moist air is brought from the ocean, that precipitate inland (Gorodetskaya et al., 2013). These regimes occur 41-48 % of the time (Gorodetskaya et al., 2013, 2014). Very few blowing snow events occur in cloudless cold conditions (cold katabatic regime), when the northerly winds blows from the interior towards the coast (N = 139; 8%).*

*Intense mixed events ( see Fig.5 ) occur together with north-easterly strong winds : 87°to N, 10 $\mathrm{m\cdot s^{-1}}$ at PE and 65°to N,*

10   *13 $\mathrm{m\cdot s^{-1}}$ at Neumayer III , warmer surface temperatures and higher relative humidity. These are the signature of storms associated with synoptic events, during which the turbulent mixing reduces the vertical temperature gradient (Gorodetskaya et al., 2013). The majority (60 %) of the blowing snow events occur during storms or overcast conditions (with cloud and/or precipitation). These mixed events have generally a short time lag since the last precipitation event and reach high atmospheric levels. Dry blowing snow has a mean wind direction of 120°to N at PE and 77°at Neumayer III, lower wind speeds (6-7$\mathrm{m\cdot s^{-1}}$)*

15   *and a greater temperature inversion at. The mean time lag since the last precipitation event at PE (23 hours) indicates that these events most likely occur after a storm, and that cloudless blowing snow (8 %) is mostly associated to katabatic winds.*

**20   Question 20: P16, Fig. 10 (caption): Indicate the relative proportion of each category.**

The proportions have been added to the Figs. 3 and 4 above (Figs.9 and 10 in the new manuscript).

**21   Question 21: P16, L4: "as", not "although".**

20   The sentence has been corrected

*a great part of the events during the synoptic regime would be missed, as they represent more than half of the events observed at PE*

**22   Question 22: P16, L13: Remove "anymore".**

25   The sentence has been corrected

*This is, however, not so obvious if we normalize the distribution of blowing snow events taking into account the total number of measurements within each time lag after precipitation.*

[Figure]

[Figure]

**Figure 6.** Determination of the height of the layer by the BSD algorithm. (a) in case of a cloud free blowing snow profile, the height of the layer is attained when the backscatter intensity reaches the clear sky threshold. (b) in case of precipitation, the height of the blowing snow layer is reached when the intensity of the backscatter signal re-increases.

**23 Question 23: P17, section 4.3.2: It is not clear how the depth of the blowing snow layer is determined.**

The explanation fo the blowing snow depth determination lies in P11, L2-7, a reference to this section as been added. In addition, illustrations are added in the Supplements (Fig.6 above, Fig. S3 in supplements)

*The height of the blowing snow layer (algorithm explained in section 3.2.) varies according to different parameters: wind speed, and the size and density of the snow particles.*

and section 3.2.

*In addition to the detection of blowing snow, the BSD algorithm quantifies the height of the layer (see Fig. S3, supplements) This is done as follows; if the profile decreases steadily (indication of absence of precipitation), the range gate at which the intensity of $\beta_{att}$ drops under the clear sky threshold value is the top of the layer. Anything above this height is considered clear sky. If there is precipitation or a cloud during the blowing snow event, the shape of the backscatter profile does not decrease*

15 *monotonously, but shows an increase in higher levels. In that case, the range gate at which the profile increases again is the top of the blowing snow layer, and the base of the cloud and/or precipitation.*

**24 Question 24: P18, Fig. 11 (ordinate axis): Indicate the units.**

The figure on page 18 is Figure 13, the figure label has been changed accordingly.

[Figure]

**Figure 7.** Scatter plot of the time since last precipitation event versus height of the blowing snow layer. Each point represents a blowing snow event. The colorbar represent the data density (number of observations divided by the entire sample size).

**25 Question 25: P18, L10: If your algorithm is applied "successfully", then you consider the visual observations as ground truth. Compare favorably with or something like that, would be more appropriate. Idem for "proved the applicability".**

Indeed, this suggests that we consider visual observations as ground truth, which is not the case. The text has been changed accordingly.

*The BSD algorithm developed for the Vaisala CL-31 ceilometer at PE was applied to the Vaisala CL-51 ceilometer at Neumayer III station. Comparing the BSD algorithm detections to visual observations at Neumayer showed a good agreement and the ability of the BSD algorithm to detect (heavy) blowing snow events, both under dry and precipitating conditions.*

**26 Question 26: P19, L15: Metamorphism does not impact the friction velocity, only the threshold friction velocity (see comment 1).**

The sentence has been changed accordingly

*These parameters change with metamorphism and impact the threshold friction velocity, and thus the and minimum wind speed required for particles uplift from the ground.*

**27 Question 27: P19, L17: Can you give more examples of such (many) studies?**

Giovinetto et al. (1992), Déry and Yau (1999), Déry and Yau (2002), Yang et al. (2010) and Palm et al. (2011).

*Here, we do not apply any wind speed threshold to the detection of blowing snow, whereas some modelling studies assume a drifting snow dependency on temperature and wind speed (Giovinetto et al., 1992; Déry and Yau, 1999, 2002; Yang et al., 2010). Palm et al. (2011) for instance, uses a minimum wind speed criterion to detect blowing snow from satellite backscatter, potentially leaving out some events.*

**28 Question 28: P19, L19: a "the" is redundant. The properties listed in brackets are not complementary information of "freshly fallen snow". Please rephrase.**

The sentence has been adapted accordingly

*We find that the presence of freshly fallen snow has a great impact on blowing snow occurrence and blowing snow layer height.*

**29 Question 29: P19, L29: Which role do you give to the turbulence during katabatic conditions in limiting the occurrence of blowing snow at PE?**

The sentence was wrongly phrased. The 'limited ' was intended to be related to availability, but also to turbulence. During the katabatic regime, there is little turbulence at PE station, as the greater temperature inversion than for the synoptic regimes suggests. Less turbulence, therefore less particles lifted from the ground.

*At PE, the explanation for the limited occurrence of blowing snow under katabatic conditions might lie in the fact that the station is shielded by the Sør Rondane mountains: wind speeds are lower and turbulence is reduced due to the very stable conditions that are frequently present (Gorodetskaya et al., 2013). In addition, the availability of fresh snow is limited as the time lag since the last precipitation event is greater, compared to synoptic conditions.*

**30    Question 30: P19, L29-31: Katabatic winds or conditions, not "katabatics". Please clarify where and how the effect of katabatic winds on the occurrence of blowing snow has been overestimated? Do you actually mean that katabatic winds are not the main driven force behind blowing snow at PE, as usually considered? If so, you should limit this conclusion to the particular geographical settings of PE, which are likely non-representative of the general conditions in coastal East Antarctica.**

Yes, The analysis of blowing snow occurrence at Princess Elisabeth station reveals that there are fewer blowing snow events during the cold katabatic regime : N = 152, 8%, than during the warm synoptic or transitional regimes. This is also illustrated in Fig.3 above: the wind roses show 1 to 2 % of blowing snow taking place during northerly winds. These special conditions at PE have been also described by Thiery et al. (2012) showing that most of the drifting snow sublimation occurs during transitional synoptic regime when the winds are strong due to the nearby cyclone, while air is undersaturated. Larger occurrences of katabatic winds are found in the absence of blowing snow. This indicates that blowing snow occurs predominantly under easterly and north easterly winds, and that the effect of katabatic winds are not the main driver for blowing snow occurrence at PE station. Regarding Neumayer III station, we find that blowing snow occurs mainly during synoptic disturbances, which is also stated by König-Langlo and Loose (2007): "blowing snow is limited to synoptic disturbances and advection from the east". Please note that we discuss significant blowing snow events (layers higher than 30 m height). Drifting snow might give different results, but is not investigated in this paper.

*At PE, the explanation for the limited occurrence of blowing snow under katabatic conditions might lie in the fact that the station is shielded by the Sør Rondane mountains, but also due to the limited availability of fresh snow and the reduced turbulence during those events compared to synoptic conditions, maintaining particles aloft. This, together with the reduced number of blowing snow events occurring under katabatic winds (Fig. 10) might indicate that the effect of katabatic winds on blowing snow occurrence has been overestimated, and that synoptic events bringing fresh snow is a most possibly determining factor for blowing snow at Neumayer III and PE stations.*

**31    Question 31: P20, L7: Specify that this conclusion is only valid for PE.**

The sentence has been adapted accordingly. However, this is also valid at Neumayer III station.

*The presence of precipitation does not substantially limit the retrieval by the ceilometer. This is an improvement to satellite detection, limited to clear sky conditions and therefore missing a great part of the blowing snow as more than half of the blowing snow happens during a storm at PE and Neumayer III station.*

**32    Question 32: P20, L9: "mainly determines".**

the sentence has been changed accordingly

5      *The availability of fresh snow mainly determines the onset of blowing snow, and the available fresh snow can be lifted to higher heights than during katabatic conditions whose effect is likely to have been overestimated for lifting snow from the surface.*

**33    Question 33: P20, L10-11: In which context this conclusion has been drawn?**

The majority of the blowing snow events occur during transitional or warm regimes at both stations (around 92 %), and only a
10   limited number of blowing snow events have been retrieved during katabatic conditions. In addition, 60 % of the blowing snow
events happen together with precipitation, indicating synoptic or transitional events rather than katabatic conditions.

*We further conclude that most of the blowing snow events happen during or shortly after precipitation, brought to the continent by the easterly winds associated to synoptic systems. The availability of fresh snow mainly determines the onset
15   of blowing snow, and the available fresh snow can be lifted to higher heights than during katabatic conditions at PE and Neumayer stations. This highlights again the limitation of wind speed thresholds, when applied to blowing snow retrieval methods. The properties of the snow particles, as well as the availability of fresh snow need to be taken into account in order to accurately initiate blowing snow in models.*

**34    Question 34: P20, L12: "The availability": you mean erodibility (availability of fresh snow is not a snow**
20         property)?

"Including" has been changed to "and".

*This highlights again the limitation of wind speed thresholds, when applied to blowing snow retrieval methods. It also emphasizes the need to take into account the properties of the snow particles and the availability of fresh snow, in order to
25   accurately initiate blowing snow in models.*

**35    Question 35: P20, L15: Use "evaluate" rather than "validate".**

the sentence has been changed accordingly

[revised manuscript text omitted]

---

## Author Comment (AC2) · 13 Sep 2017

**Response to Reviewer 2 Comments:**

**Blowing snow detection from ground-based ceilometers: application to East Antarctica**

Alexandra Gossart[1], Niels Souverijns[1], Irina V. Gorodetskaya[2,1], Stef Lhermitte[3,1], Jan T.M. Lenaerts[4,1,5], Jan H. Schween[6], Alexander Mangold[7], Quentin Laffineur[7], and Nicole P.M. van Lipzig[1]

[1]Department of Earth and Environmental Sciences, KU Leuven, Leuven, Belgium
[2]Centre for Environmental and Marine Sciences, Department of Physics, University of Aveiro, Aveiro, Portugal
[3]Department of Geosciences and Remote Sensing, Delft University of Technology, Delft, the Netherlands
[4]Institute for Marine and Atmospheric research Utrecht, Utrecht University, Utrecht, The Netherlands
[5]Departement of Atmospheric and Oceanic Sciences, University of Colorado, Boulder CO, USA
[6]Institute of Geophysics and Meteorology, Koeln University, Koeln, Germany
[7]Royal Meteorological Institute of Belgium, Brussels, Belgium

For clarifying our answers to the referees' comments, the following scheme is used: comments of the referees are denoted in **bold**, our answers are denoted in black and quotes from the revised text are in *italic*. Please note that reference to figures in the answer refer to the original manuscript, or to the improved figure displayed in the Response document. Figures referenced in the italic text are relative to the new manuscript.

**General comments: The calibration of the ceilometer remains an issue and I wonder if it will be possible. This is important since the use of a ceilometer would start to be relevant for blowing snow studies if a quantitative retrieving of blowing snow characteristics (height of the blowing snow layer, amount of transported snow) may be done. Is it the intention of the authors to perform such a calibration in the future? This point must be considered in the discussion and**

10 **the conclusion in order to advice the reader about the potentialities and weaknesses of the study. Nevertheless the paper represents a sufficient amount of work to be published. To my opinion the technical description should be improved for a TC reader, especially a modeller. In fact there is too much or not enough. An alternate possibility should be to shorten the technical description and to do it in an other more specialized journal. I had difficulties with a double meaning of some sentences (see specific comments below).**

15

Thank you for your comments, we have replied to each of the comments below.

Calibration of the ceilometer to quantitatively retrieve the amount of transported snow is indeed an issue, as this can not be derived from the ceilometer attenuated backscatter signal. With the current instrumentation this is not possible. Furthermore, we derived a blowing snow algorithm for instruments already present at Princess Elisabeth station. Lidars can be used to define

20 the lidar ratio, but these instruments are (1) more expensive and (2) less abundant than ceilometers. Even after ceilometer

calibration, the amount of transported snow can not be derived from particles properties only. We would require to estimate the transport rate also.

*We present here our novel BSD algorithm, designed to retrieve blowing snow events, but not drifting snow, from ground-based remote-sensing ceilometers. Ceilometers can retrieve the presence of blowing snow, but other properties such as size,* 5 *shape and density measurement is only possible if the ceilometer is calibrated, which is very challenging for such a remote location, and not done in this paper.*

The algorithm has been adapted to derive precipitation/cloud occurrences from the ceilometer profile directly, and the new version of the paper contains an improved technical description of the algorithm (containing the bin numbers and threshold 10 values, text below), together with a scheme of the concept of the blowing snow algorithm (Fig.1 below, Fig. 5 in the new manuscript).

[revised manuscript text omitted]

**1 Question 1: p. 2, line 5. : the word "suspension" is defined here but is no used in the rest of the paper (see e.g., line 18 p.7), so that its introduction here is not clear.**

Indeed. I have re-worked this part of the introduction.

10 *This phenomenon occurs approximatively on 70 % of the Antarctic continent during winter (Palm et al., 2011) and snow is transported as "drifting snow" ( if the vertical extend of the layer is lower than 2 m), or as "blowing snow" (layers more than 2*

m *height). These transport involve a mix of suspension and saltation transport modes (Leonard et al., 2011), with a dominance of saltating particles (Bagnold, 1974) in the case of drifting snow, and suspended particles in blowing snow layers (Mellor, 1965).*

**2 Question 2: p.2, line 24, note that the precipitation process is also poorly constrained in Antarctica so that the authors have to face to one equation on SMB with at least two unknowns: precipitation and snow erosion by the wind.**

Indeed, although there are products available such as stake measurements (SAMBA dataset (Favier et al., 2013), and observations from the Cloudsat satellite (Palerme et al., 2014)) which allow precipitation estimates on large areas over the continent. In addition, at the Princess Elisabeth station, we have a micro-rain radar that enables to measure precipitation rates. Erosion by the wind is much more difficult to predict there, and is treated as a residual term, containing all the uncertainties on the other terms.

*Currently, simulations of the AIS SMB are highly uncertain since both precipitation and blowing snow processes are poorly constrained and probably lead to inconsistencies between the atmospheric modeled precipitations and the measured snow accumulation value (Frezzotti et al., 2004; Scarchili et al., 2010; Groot Zwaaftink et al., 2013; Gorodetskaya et al., 2015; van de Berg et al., 2005).*

**3 Question 3: p.3, section 2. What are the altitude of both stations PE and Neumayer. Is their climate (e.g., SMB, summer temperature, ...) different? This will help the reader when considering the development of the BSD by using observations at Neumayer and using it for another location.**

A table has been added in the new manuscript (Table 2), presenting the climate at both stations (see Table 1 below). PE station is located on Utsteinen ridge, 1392 m a.s.l. and 173 km inland. Neumayer station is located on the ice shelf at 43 m a.s.l. Their climate is indeed different. Neumayer is subject to higher wind speeds ($9 \ \mathrm{m \cdot s^{-1}}$) than PE station ($5 \ \mathrm{m \cdot s^{-1}}$) and higher relative humidity. PE is located further from the ocean, and is shielded from the katabatic winds by the Sør Rondane mountains. Accumulation is lower due to the distance to the coast. Surface temperature are similar, around -16 / -17 °C. For extended information on Neumayer III and PE meteorology, see König-Langlo and Loose (2007) and Gorodetskaya et al. (2013).

**4 Question 4: p.3, section 2.1. An introductory sentence stating that the ceilometer was not initially set up for measuring blown snow events would clarify the section. More generally the description here should contain more information related to a possible use of the measurements for a determination of blown snow characteristics.**

Indeed, although it is already stated at the end of the introduction (previous section), the paragrah has been adapted accordingly. The ceilometer measurement can not be used to determine anything else than blowing snow occurrence. Quantification

**Table 1.** Climatic conditions at Princess Elisabeth , and Neumayer III stations. For extended climatology, see Gorodetskaya et al. (2013) for PE station and König-Langlo and Loose (2007) for Neumayer station.

| variable | Princess Elisabeth | Neumayer III |
| --- | --- | --- |
| coordinates | 71 °57' S; 23 °21' E | 71 °56' S; 23 °20' E |
| distance from the coast | 173 km | approx. 7 km |
| elevation | 1392 m asl | 43 m asl |
| average air temperature | -18 °C | -16 °C |
| average wind speed | $5 \, \text{m} \cdot \text{s}^{-1}$ | $9 \, \text{m} \cdot \text{s}^{-1}$ |
| average wind direction | | |
| • synoptic disturbances | 90 °to N | 100 °to N |
| • katabatic conditions | 180 °to N | 170 °to N |
| relative humidity | 56 % | 90 % |
| pressure | 827 hPa | 986.5 hPa |

of blowing snow displacement, and the determination of blowing snow properties such as particles density, shape or number can not be derived from the ceilometer attenuated backscatter signal.

*Initially set up to measure cloud base height, ceilometers are rather simple and robust instruments. The algorithm described*
5 *in this paper was built to derive blowing snow occurrence from the signal received by these devices.*

and section 2.1.

*The quantitative information that can be derived from the ceilometer measurements, is the attenuated backscatter intensity*
10 *at defined heights (Wiegner et al., 2014; Madonna et al., 2015). Other properties such as optical depth, size and density would require to know the lidar ratio,and a reliable estimate of lidar ratio is complicated (Wiegner et al., 2014). In addition, this is only possible if the ceilometer is calibrated, which is very challenging since the signal to noise ratio has to be large enough in the troposphere (Wiegner et al., 2014) and is not done in the present study. This implies that quantification of blowing snow displacement, and the determination of blowing snow properties such as particles density, shape or number can not be derived*
15 *from the ceilometer attenuated backscatter signal at Neumayer III and PE stations*

**5 Question 5: p.4, line 1. : what is the raw resolution in time of the ceilometer?**

The reporting invterval is of 2 s. This is stated in Table 1 in the original manuscript, and the sentence has been removed, for clarity.

**6 Question 6: p.4, line 2. : "spatial resolution": do you mean "vertical"?**

Yes indeed. The sentence has been adapted accordingly

*The ceilometer measures continuously and the standard output, $\beta_{att}$ is displayed in a time-height cross section, with a 10m*
5 *vertical resolution and 15 s temporal resolution.*

**7 Question 7: p.4, line 20. : please indicate for each instrument which measurement you intend to use in the paper, especially concerning the infrared pyrometer (see also p.6, line 2 where it is not said there for which purpose the cloud base height deduced from the brighness temperature is used). As for the next comment it is preferable to describe the use of an instrument in a single paragraph.**

10 The cloud base temperature is used as a near-atmospheric variable. In the case of blowing snow, the measured cloud tempera-ture is actually the blowing snow layer temperature. It was used in the cluster analysis, and in the PCA. However, it was not a determining variable. This information is left out in the new version of the paper. Only the micro-rain-radar is used to retrieve precipitation rates, in addition to the meteorological variables measured by the automatic weather station.

15 *The Metek vertically-profiling precipitation radar, set up since 2010, enables to retrieve snowfall rates, using the return from the vertically profiling Doppler radar operating at a frequency of 24 GHz.*

**8 Question 8: p. 5, lines 4-6. The ceilometer is described twice. Please rearrange the text.**

The text has been rearranged accordingly.

20 *A cloud and precipitation observatory was set up on the roof of the station (approx. 10 m above the ridge) during the sum-mer season of 2009-2010 and is still operational under the Hydrant/Aerocloud project (www.aerocloud.be). The observatory contains an automatic weather station (AWS) and a set of ground-based remote sensing instruments. The observatory was designed to be operated year-round, including the winter period when PE is unmanned. The station and the set of instruments are controlled remotely via a satellite connection.*
25 *The Vaisala CL-31 ceilometer (firmware 1.72) was installed on the roof of the station in December 2009 and is operational at present. It emits laser pulses at central wavelength of $910 \pm 10$ nm at 298 K. The measurement vertical resolution is set to 10 m and the reporting interval on 15 s. Several outages of the energy provision system limit the data mainly to Antarctic summer season (December to March is best represented). Only one year of continuous measurements was achieved (2015).*
*The Metek vertically-profiling precipitation radar, set up since 2010, enables to retrieve snowfall rates, using the return from*
30 *the vertically profiling Doppler radar operating at a frequency of 24 GHz. The raw Doppler spectra is post-processed follow-ing Maahn and Kollias (2012), to calculate radar reflectivity profiles which are then linked to snowfall rates using the newly*

*developed Ze-Sr relation for PE by Souverijns et al. (2017) and has a sensitivity up to -14 and -8 dBz (Souverijns et al., 2017).*
*A full description of micro-rain radars can be found in Klugmann et al. (1996) and the radar set up at Princess Elisabeth is*
*described in Gorodetskaya et al. (2015).*

*The monitoring of the instruments set up on the roof of the station is done via a webcam. Specifications of the instruments are*
5 *given in Table 2 (see also Gorodetskaya et al. (2013, 2015)).*

**9 Question 9: p.5, lines 8-9. Please clarify the description of the MRR.**

The description of the MRR has been adapted, and references to MRR description (Klugmann et al., 1996) and the specific radar set up at PE (Gorodetskaya et al., 2013) have been provided.

10 *see question 8 above.*

**10 Question 10: p.8, line 4. Please indicate the reason of the turning on/off of the heater.**

The heater is used to stabilize the laser temperature in cold environments (Kotthaus et al., 2016). The heater is turned on until the device attains the temperature, then is switched off. The temperature of the instrument decreases then, due to the cold surroundings, and when a minimum temperature is reached, the heater is turned on again.

15

*There are two sources of noise and artifacts affecting the ceilometer backscatter signal: the hardware of the Vaisala ceilome-*
*ters, and the internal processing of the data (Kotthaus et al., 2016). Firstly, a heater is incorporated in the device to stabilize*
*the laser temperature in cold environments. This heater is placed close to the laser transmitter and the periodic turning on*
*(when a minimum temperature is reached by the instrument) and off (when the laser temperature is high enough) of the heater*
20 *introduces a small periodic variation in the stability of the emitted signal (and therefore of the detected signal). This effect is*
*stronger in the first range bins, closest to the device.*

**11 Question 11: p.14, line 13. What about the role of sastrugi in the evolution of blowing snow intensity?**

Indeed, the presence of sastrugis has an impact on blowing snow intensity evolution. However, in this section we investigate the changes in near-surface atmospheric variables during blowing snow conditions. Despite their possible impact, sastrugis are
25 not measured either at PE nor at Neumayer III station.

*Apart from these factors, sastrugis might also have an impact on blowing snow (Amory et al., 2017) but are not measured*
*here.*

**12 Question 12: p.14 – 15, fig. 8 and 9. How do you quantify from a statisticall point of view the differences between blowing snow and non blowing snow wind speed and relative humidity? What is your interpretation of the differences for the other variables?**

By means of a t-test significant at the 95 % level. The difference for the other variables is not significant, meaning that blowing snow or non blowing snow conditions give similar distributions for these variables. However, due to the comments received on this section, it has been removed together with Figs. 8 and 9.

**13 Question 13: p.16, line 2. What is the advantage of satellite detection?**

The advantage of satellite detection is the spatial coverage of blowing snow. This enables Palm et al. (2011) to produce a map of blowing snow frequencies over the whole of the Antarctic continent. A sentence has been added, and the paragraph has been moved to section 5.1 (discussion).

*Satellite detections of blowing snow, although covering the whole continent, are limited to clear sky conditions. The BSD algorithm, however, is able to detect blowing snow during most of the storms, which is an improvement compared to satellite detection, as the majority of blowing snow occur together with cloud/precipitation.*

**14 Question 14: p.16, line 14. Clarify "observations"**

Observations referred to the number of measurements ; i.e. the number of times during the measurement period, that a certain time lag after precipitation is reached.

*A possible explanation is that the number of measurements decreases with time, and that blowing snow occurred during those measurements.*

**15 Question 15: p.18, lines 17 – 18: "commission errors": please clarify.**

"Commission error" was stated twice, and should only appear once. The second mention should have been "ommission error". In our case, a commission error is a BSD detection that is not reported by the visual observer. It is similar to a "false alarm", but since we do not consider visual observations as ground truth, but as another means of measuring blowing snow, we chose the omission/commission terms. The omission error refers to missing a blowing snow occurrence that is reported by the visual observer.

*Furthermore, the hourly time filtering applied leads to commission errors (events detected by the algorithm, but not reported by the visual observations) and ommission errors (short-lived events are likely removed from the running mean).*

**16   Question 16: p.19, line 2. Is it possible to improve the set-up of the ceilometers on the field, and how?**

Indeed, however, most of the ceilometers are intended to forecast the weather for planes landing. Depending on the purpose
of ceilometer measurements, the ceilometer could be placed closer to the ground to measure lower level blowing snow, and reporting resolution can be adapted (10 m vertical resolution).

*If setting up a ceilometer in the aim of measuring blowing snow, the device should be placed as close to the ground as possible to also retrieve shallower blowing snow events. The BSD algorithm can be applied to any ceilometer located in Antarctica, but we recommend to use a bin width of 10 m for operating ceilometers to detect blowing snow, which is the case at PE and Neumayer III.*

**17   Question 17: p.20, line 3. : ... designed to retrieve blowing snow events but no drifting snow from ground-based ...**

The sentence has been adapted accordingly.

[revised manuscript text omitted]

---

## Author Comment (AC3) · 13 Sep 2017

**Response to Reviewer 3 Comments:**

**Blowing snow detection from ground-based ceilometers: application to East Antarctica**

Alexandra Gossart[1], Niels Souverijns[1], Irina V. Gorodetskaya[2,1], Stef Lhermitte[3,1], Jan T.M. Lenaerts[4,1,5], Jan H. Schween[6], Alexander Mangold[7], Quentin Laffineur[7], and Nicole P.M. van Lipzig[1]

[1]Department of Earth and Environmental Sciences, KU Leuven, Leuven, Belgium
[2]Centre for Environmental and Marine Sciences, Department of Physics, University of Aveiro, Aveiro, Portugal
[3]Department of Geosciences and Remote Sensing, Delft University of Technology, Delft, the Netherlands
[4]Institute for Marine and Atmospheric research Utrecht, Utrecht University, Utrecht, The Netherlands
[5]Departement of Atmospheric and Oceanic Sciences, University of Colorado, Boulder CO, USA
[6]Institute of Geophysics and Meteorology, Koeln University, Koeln, Germany
[7]Royal Meteorological Institute of Belgium, Brussels, Belgium

For clarifying our answers to the referees' comments, the following scheme is used: comments of the referees are denoted in **bold**, our answers are denoted in black and quotes from the revised text are in *italic*. Please note that reference to figures in the answer refer to the original manuscript, or to the improved figure displayed in the Response document. Figures referenced in the italic text are relative to the new manuscript.

**General comments:**

**1. The evaluation of the performance of the detection algorithm is based on the comparison with human observations at Neumayer III. The different classes (occurrence or not) may be unbalanced (much more cases without blowing snow than with, as suggested on l.14, p.13) requires more robust statistics than the one used. There is a lot of literature about**

10 **what criteria can be employed for such confusion matrices. See for instance Allouche et al. (2006). I suggest the authors to sue such commonly used statistics (e.g. Cohen's kappa, true skill statistics) for the evaluation of the algorithm. The estimated depth is not really evaluated, what would be needed to do so.**

Thank you for this excellent remark. I have used the statistics indicated in Allouche et al. (2006) (see answer to Question 6

15 below). Regarding the depth of the layer measured by the ceilometer, it is likely underestimated, due to the attenuation of the signal. A way to evaluate the layer height, is to compare height measured by the BSD to satellite detections (data from Palm et al. (2011)), when concurrent. This is part of an ongoing work and not included in this paper.

**2. The statistics derived from the outcome of the blowing snow detection algorithm are informative and relevant, but**

20 **they could be more complete, by including data and analysis about the inter- and intra-event variability of the blowing**

[Figure]

**Figure 1.** Inter- and intra variability of blowing snow layer height. Each color represents a distinct blowing snow event.

**snow occurrence and depth.**

We have performed this analysis on the blowing snow depth, and time versus last precipitation (Figs. 11 and 13 in the original manuscript), but the results showed no real inter variability between the events: Fig.11 was similar. The intra-variability in the layer height versus time since last precipitation (Fig. 1 above) shows two types of events: a majority of blowing snow layers of stable height, and a few events display an increase/decrease in layer heights. Given the limited amount of data and the focus of the paper, we decided to leave this out of the manuscript.

**1   Question 1: P.7, l.28: the choice of smoothing the signal over 1 h should be better justified (why 1 h and not 30 min or 2 h?). The typical variability of the BS layer features should be commented (if there is a lot of dynamics within 1 h, one may loose relevant information by smoothing over 1 h).**

The reason the running mean was set to 1 hour was to (1) smooth out effects of turbulence, but (2) mainly to get rid of the periodic fluctuation in the signal due to the heater switching on and off. 30 mins is not enough to smooth this out (see figure 2 below), and one hour was chosen since it is the lowest time period at which the heater artifact was substantially smoothed out. This was indeed not mentioned in the original manuscript.

*We average every 15s- $\beta_{att}$ profile over one hour using a running mean, to create mean attenuated backscatter profiles at every time step and avoid the variability due to turbulence and hardware noise. Figure 4. shows the resulting $\beta_{att}$ at 09:30 UTC, based on the average of 240 profiles (120 preceding and 120 following 09:30 UTC). An additional reason for the integration of the signal over longer time periods, is that it improves the signal to noise ratio (SNR). No additional SNR correction is*

[Figure]

**Figure 2.** analysis of the periodic fluctuation visible in the ceilometer signal. Second range bin, 06.02.2013, clear sky day.

*performed on the raw data, as we found that a temporal SNR higher than 0.3 would remove parts of the blowing snow signal (Gorodetskaya et al., 2015).*

*There are two sources of noise and artifacts affecting the ceilometer backscatter signal: the hardware of the Vaisala ceilometers, and the internal processing of the data (Kotthaus et al., 2016). Firstly, a heater is incorporated in the device to stabilize the*
5 *laser temperature in cold environments. This heater is placed close to the laser transmitter and the periodic turning on (when a minimum temperature is reached by the instrument) and off (when the laser temperature is high enough) of the heater introduces a small periodic variation in the stability of the emitted signal (and therefore of the detected signal). This effect is stronger in the first range bins, closest to the device, and the hourly running mean enables to smooth out most of this signal variation.*

10 ## 2 Question 2:P.8, l.1: "SNR higher than 0.3": I guess it is expressed in dB. If so, it should be clearly mentioned.

The SNR is the signal-to-noise ratio, and is uniteless. It is calculated at each height range bin *j* at time step *i* as :

$$SNR_{i,j} = \frac{\overline{\beta_{i,j}}}{\sqrt{\frac{1}{2M}\sum_{k=-M}^{+M}(\beta_{i+k,j} - \bar{\beta_{i,j}})^2}} \tag{1}$$

which is the ratio of the temporal mean $\bar{\beta}_{i,j}$ and standard deviation of the attenuated backscatter over $\pm M$ time steps around time step $i$ and range bin $j$ (Van Tricht et al., 2014).

*No additional SNR correction is performed on the raw data, as we found that a temporal SNR higher than 0.3 would remove*
5   *parts of the blowing snow signal (Gorodetskaya et al., 2015).*

**3   Question 3: P.10, Fig.5: I probably missed something, but I do not understand why the backscatter signal from BS+precip is so much smaller (below 100 m alt) than the one from BS only (red vs blue). I would expect the two signals to sum up somehow... Or is the concentration in BS particles much smaller when there is precip? If so, what could be the explanations?**

10   Figure 5 (Fig. 6 in the new manuscript) is indeed not clear, as we chose one day (24.04.2016) where the different typical events occur. During that day, blowing snow accompanied with cloud/precipitation unfortunately occurred at the end of the blowing snow event. Hence, the lower intensities and the resemblance to the pure precipitation profile. We have therefore selected an-other day (10.02.2014) to illustrate the pure precipitation, and the mixed event. In this new figure (Fig.3 below, Fig.6 in the new manuscript), the intensity of the profile in the lowermost bins is clearly indicative of blowing snow (red line), and the increase
15   around the 15th bin indicates the presence of clouds/precipitations (arrow). In the case of precipitation/cloud without blowing snow, this low-level decrease is absent, and the increase around the 17th bin reflects the presence of a cloud/precipitation.

**4   Question 4: P.10, l.19: related question: it is written "The precipitation intensity might cover the blowing snow signal", which I find confusing with the curves in Fig.5 (for the lower altitudes). To be clarified...**

We know that most of the time, blowing snow happens together with storms and intense precipitation (the snowflakes rebound
20   on the ground and are displaced by strong winds). Hence, in some cases the signal intensity is not decreasing with height, and the profile criterion is not met. In those cases, blowing snow during very intense events was discarded by the algorithm. Fig 4 (below) presents a case of blowing snow around 23:00 UTC (intense coloration). From the profiles for 23:00 UTC it is clear that the signal of the cloud eclipses that of blowing snow, even though the threshold is largely exceeded in the second bin, and the decrease in bins 2-3 is visible. As the intensity of the signal in bins 3 to 7 is larger than in the second bin, the BSD
25   algorithm does not detect blowing snow. We therefore adapted the method, applied in the new version of the manuscript. This improved algorithm limits the decrease of the profile to cases where the precipitation signal is not intense. In cases of signal in the second bin exceeding $1000 \cdot 10^{-5} \cdot \mathrm{km}^{-1} \cdot \mathrm{sr}^{-1}$ (threshold adapted from (Gorodetskaya et al., 2015)), the algorithm reports a heavy precipitation event mixed with blowing snow in all cases. A chart presenting the blowing snow detection algorithm has been added to the paper (Fig. 5 below, Fig.5 in the new manuscript).

30

[Figure]

**Figure 3.** Different types of profiles relevant for blowing snow measured by the ceilometer at PE station: blue line - typical blowing snow signal with no precipitation nor clouds (24-04-2016); red line - blowing snow overlaid by precipitation (10-02-2016); black line - precipitation with no blowing snow (10-02-2014); yellow line - near-zero signal for clear sky conditions (24-04-2016). The height above ground is indicated on the right axis and the corresponding bin number on the left axis. All profiles exclude the lowermost bin, and start at the second bin (15 m agl.). The grey lines represent the discontinuity between bins 4 and 5 (35-45 m). The arrows indicate the presence of precipitation.

*The algorithm therefore investigates the shape of the profile in order to detect blowing snow. A condition is set, that a blowing snow profile implies that the mean of the overlying bins 3 to 7 (25 to 65* m*) must be lower than the signal in the second range bin (15* m*). In this way, the discontinuity, as described in section 3.1. (visible in Figures 1 and 6 between 35 and 45 m), is not affecting our retrievals. In order to detect blowing snow occurring during clouds or precipitation, the profile shape is analyzed*

5 *to identify a second increases in the signal intensity above the 7th bin (65 m height). A clear differentiation between clouds or precipitation cannot be made on the basis of the ceilometer alone, but the presence of clouds and/or precipitation can be identified. This analysis is carried out for both blowing snow and non blowing snow measurements. The information retrieved from the Micro Rain Radar (hourly precipitation rates) is collocated to ceilometer blowing snow detection, to determine the time (in hours) since the last precipitation event at PE station.*

10 *Inherent to this profile-based method, the detection of blowing snow during precipitating events is limited to cases when the blowing snow signal is preserved close to the ground. In case of precipitation associated with storms, there is always blowing snow due to the high wind displacing the snow, and no distinction between precipitation and blowing snow is possible, as the ceilometer signal is entirely attenuated near the surface (Gorodetskaya et al., 2015), it is not possible to get signal in*

[Figure]

**Figure 4.** Quicklook presenting a mixed blowing snow event : clouds and precipitation occurring together with blowing snow around 23:00 UTC (left), and ceilometer attenuated backscatter profiles (intensity versus height) around 23:00 UTC (right).

[Figure]

**Figure 5.** Chart of the blowing snow detection method

*the overlying bins, and the profile of the backscatter intensity might not decrease upwards. Such intense precipitating events mixed with snowfall are identified as having a second bin signal higher than $1000 \cdot 10^{-5} \cdot km^{-1} \cdot sr^{-1}$ (threshold adapted from Gorodetskaya et al. (2015)). In those cases, the events are classified as a mixed blowing snow event, and the profile analysis is eluded by the algorithm.*

[Figure]

[Figure]

**Figure 6.** Determination of the height of the layer by the BSD algorithm. (a) in case of a clear sky blowing snow profile, the height of the layer is attained when the backscatter intensity reaches the clear sky threshold. (b) in case of precipitation, the height of the blowing snow layer is reached when the intensity of the backscatter signal re-increases.

**5 Question 5: P.11, l.6-7: about the estimation of the top of the BS layer: it would help the reader to indicate in Fig.5 where is this limit. And how reliable would be the outcome in case of virgas?**

Indeed. The graphs have been added (Fig. 6 above) to the Supplements (Fig. S3 in the new manuscript)

5    In our case, no distinction is made between clouds, precipitation and virga. this means that any re-increase in the profile is treated as the presence of clouds and/or precipitation.

*If there is precipitation or a cloud/virga during the blowing snow event, the range gate at which the profile increases again is the top of the blowing snow layer, and the base of the cloud and/or precipitation layer (around the 7th bin in Fig.5, for the*
10  *black and the red profiles). Layer height definition is illustrated in Fig. S3 in the supplements.*

**6 Question 6: P.12, l.7-14: better metrics could be computed to evaluate the performance of the algorithm, see General comments above.**

Indeed, thank you for this excellent remark. I have used the statitics indicated in Allouche et al. (2006).

[revised manuscript text omitted]

**7   Question 7: P.14, l.3: Which statistical tests have been used to check if there are "statistically significant differences"?**

By means of a t-test, at the 95 % significance level. The difference for the other variables is not significant, meaning that blowing snow or non blowing snow conditions give similar distributions for these variables. However, due to the comments received, this section has been removed from the new version of the manuscript, together with Figs. 8 and 9.

**8   Question 8: P.15, l.16: a minimum of description should be provided about the clustering method employed, so the reader does not have to check the reference to know what type of clustering method has been used for instance...**

Some information has been added. However, since the ceilometer profile classification is also useful to discriminate between the different types of events, less emphasis is put on the cluster analysis.

*The near surface atmosphere changes, associated with blowing snow events, are investigated for both stations, and detailed means and standard deviation are displayed in Table S6 and S7, in supplements. We investigate how blowing snow hourly means relate to weather regimes, derived from the hierarchical cluster analysis using PE AWS data following Gorodetskaya et al. (2013), which defines the weather regimes at PE station: "cold katabatic", "warm synoptic", and "transitional synoptic". The cold katabatic regime is characterized by slower wind speeds and lower humidity, reduced incoming long wave radiation, a slight surface pressure increase, and a substantial temperature inversion. Warm synoptic conditions involve higher wind speeds and specific humidity, strongly positive anomalies of incoming long wave radiation. The surface pressure is slightly lower, and the temperature inversion is strongly reduced than during average conditions. Finally, average wind speeds, humidity and incoming long wave radiation, as well as slightly lower surface pressure are observed during the transitional regime, when the situation evolves from synoptic to katabatic or the other way around (Gorodetskaya et al., 2013).*

**9   Question 9: P.16, Fig.10: the font size of the text in the figure should be increased.**

The font size has been adapted accordingly (see Fig.7 below).

**10   Question 10: P.16, l.14: I do not understand why the number of observations would decrease... The ceilometer is collecting data every 15s, no? The explanation should be clarified.**

The sentence was wrongly phrased. "Observations" has been replaced by "measurements". The number of "observations/measurements" decreases because it is rare that there is no precipitation for a dozen of days continuously.

[Figure]

**Figure 7.** Wind roses presenting the wind direction for all blowing snow, blowing snow with or without precipitation , and non-blowing snow conditions at Princess Elisabeth station.

*This is, however, not so obvious if we normalize the distribution of blowing snow events taking into account the total number of measurements within each time lag after precipitation [...]. A possible explanation is that there are less measurements as we go in time, and that blowing snow occurred during those measurements.*

**11 Question 11: P.18, Fig.13: are these distributions for the two stations (Neumayer and PE) or only one location?**

5    This is for PE station only, since the time since the last precipitation event is only available using the micro-rain radar. The text and legend have been adapted accordingly.

*We further tested the hypothesis that the height of the blowing snow layer is related to wind speed at PE station. While there is no correlation, (also found by Mahesh et al. (2003)), the height of the blowing snow layer is related to the time since last*
10    *precipitation (Fig. 13). The height of the blowing snow layer can reach up to 1000 m within 24 to 48 h after precipitation, and 95 % of the blowing snow layers thicker than 500 m occur shortly after the last precipitation event at PE. Blowing snow events taking place much later after the precipitation event are limited to a vertical extend lower than 100 m thick.*

and legend :

*Scatter plot of the time since last precipitation event versus height of the blowing snow layer at PE station. Each point represents a blowing snow event. The colorbar represent the data density (number of observations divided by the entire sample size).*

**12    Question 12. P.18, l.17: "commission errors" is repeated twice.**

"Commission error" was stated twice, and should only appear once. The second mention should have been "ommission error". In our case, a commission error is a BSD detection that is not reported by the visual observer. It is similar to a "false alarm", but since we do not consider visual observations as ground truth, but as another means of measuring blowing snow, we chose the omission/commission terms. The omission error refers to missing a blowing snow occurrence that is reported by the visual observer.

*Furthermore, the hourly time filtering applied leads to commission errors (events detected by the algorithm, but not reported by the visual observations) and ommission errors (short-lived events are likely removed from the running mean).*

**13    Question 13. P.19, l.9: I guess the same algorithm could be applied to lidar systems, no?**

Indeed, the algorithm could be adapted to lidars. However, we developed the algorithm for existing instruments at PE station, which is not equipped with a lidar. Moreover, such instruments are more expensive, and less widespread as ceilometers (low-cost networks).

---

## Referee Report (RR1)

According to my previous recommendation, the paper has been reorganized, clarified and valuable results are now presented in a proper way. In general, the authors have responded to my questions quite well. I think the paper warrants publication now. I only have some minor and final suggestions.

1. p. 1, line 7 (and elsewhere): The period of observation for PE station is indicated as 2010-2017. This should rather be 2010-2016?

2. p. 2, line 9: remove "and surface melt".

3. p. 16, line 15 and 18: specify to which humidity (specific or relative) you refer to.

---

## Referee Report (RR2)

Review of the revised version of the paper "Blowing snow detection from ground-based ceilometers: application to East Antarctica" by Gossart et al.

The paper has been significantly improved and I have now only minor comments which are listed below:
p.1, line 14: 1300 m a.g.l.?
p.2, line 5: suppress "Generally" (see your definition on p.7 line 13)
p.6, line 4: mention "MRR" earlier in the paragraph (e.g., on line 1)
p.8, line 2: which date?
p.13, line 8: … (referred to as ommission error)
p.13, line 27: define commission error here (see your line 2 p.20)
p.17, line13: "Blowing ...". Sentence is unclear
p.20, last line. "effect of katabatic winds on blowing snow ...". Strong katabatic winds may occur in the presence of strong synoptic scale winds in Adélie Land and are associated to blowing snow.  Please specify that your statement concerns PE and Neumayer.

---

## Author Response (AR2)

**Response to Reviewers Comments:**

**Blowing snow detection from ground-based ceilometers: application to East Antarctica**

Alexandra Gossart[1], Niels Souverijns[1], Irina V. Gorodetskaya[2,1], Stef Lhermitte[3,1], Jan T.M. Lenaerts[4,1,5], Jan H. Schween[6], Alexander Mangold[7], Quentin Laffineur[7], and Nicole P.M. van Lipzig[1]

[1]Department of Earth and Environmental Sciences, KU Leuven, Leuven, Belgium
[2]Centre for Environmental and Marine Sciences, Department of Physics, University of Aveiro, Aveiro, Portugal
[3]Department of Geosciences and Remote Sensing, Delft University of Technology, Delft, the Netherlands
[4]Institute for Marine and Atmospheric research Utrecht, Utrecht University, Utrecht, The Netherlands
[5]Departement of Atmospheric and Oceanic Sciences, University of Colorado, Boulder CO, USA
[6]Institute of Geophysics and Meteorology, Koeln University, Koeln, Germany
[7]Royal Meteorological Institute of Belgium, Brussels, Belgium

For clarifying our answers to the referees' comments, the following scheme is used: comments of the referees are denoted in **bold**, our answers are denoted in black and quotes from the revised text are in *italic*. Please note that reference to figures in the answer refer to the original manuscript, or to the improved figure displayed in the Response document. Figures referenced in the italic text are relative to the new manuscript.

Contents :

**REVIEWER 1**

**General comments : According to my previous recommendation, the paper has been reorganized, clarified and valuable results are now presented in a proper way. In general, the authors have responded to my questions quite well. I think the paper warrants publication now. I only have some minor and final suggestions.**

**1   Question 1: p. 1, line 7 (and elsewhere): The period of observation for PE station is indicated as 2010- 2017. This should rather be 2010-2016?**

Indeed, there were no measurements between may 2016 and 12.11.2016. The ceilometer was turned on at that date, to monitor the landing conditions. All the other instruments were down. I have used the few clear sky additional data, available for this

10   summer 2016/2017 for the clear sky threshold calculation. The rest of the analysis was performed on data until may 2016 only, and should appear so in the manuscript. I have adapted the dates accordingly.

**2   Question 2: p. 2, line 9: remove "and surface melt".**

" and surface melt" has been removed.

*Despite its importance, the role of blowing snow on AIS SMB is currently poorly quantified.*

**3   Question 3:p.16, line 15 and 18: specify to which humidity (specific or relative) you refer to.**

Relative humidity. "Relative" has been added.

20   *The cold katabatic regime is characterized by slower wind speeds and lower relative humidity, reduced incoming long wave radiation, a slight surface pressure increase, and a substantial temperature inversion.*

**REVIEWER 2**

**General comments: The paper has been significantly improved and I have now only minor comments which are listed below:**

**4    Question 1:p.1, line 14: 1300m a.g.l. ?**

Indeed, up to 1300 m a.g.l.

*Blowing snow occurs predominantly during storms and overcast conditions, shortly after precipitation events, and can reach up to 1300 m a.g.l. in case of heavy mixed events (precipitation and blowing snow together).*

**5    Question 2: p.2, line 5: suppress "Generally" (see your definition on p.7 line 13)**

"Generally" has been removed.

*Drifting snow events are shallower than blowing snow events.*

**6    Question 3: p.6, line 4: mention "MRR" earlier in the paragraph (e.g., on line 1)**

"MRR" is now mentioned earlier (line 1).

*The Metek vertically-profiling precipitation radar (micro-rain radar, MRR), set up since 2010, enables to retrieve snowfall rates, using the return from the vertically profiling Doppler radar operating at a frequency of 24 GHz.*

**7    Question 4: p.8, line 2: which date?**

Indeed, I should have mentioned the date. The day is 24.04.2016.

*Figure 4 shows the resulting $\beta_{att}$ for the 24.04.2016 at 09:30 UTC, based on the average of 240 profiles (120 preceding and 120 following 09:30 UTC).*
and in the caption:
*Hourly averaging of the attenuated backscatter profile of the CL-31 at PE on 24.04.2016.*

**8 Question 5: p.13, line 8: ... (referred to as ommission error)**

The sentence has been adapted accordingly.

*Mismatches occur when only one of the methods detects blowing snow, when the other does not : N $BS_{ceilo}$ if blowing snow is only reported by the BSD algorithm (commission error), and N $BS_{vis}$ when only the visual observer records blowing snow, referred to as ommission error.*

**9 Question 6: p.13, line 27: define commission error here (see your line 2 p.20)**

The definition has been moved from page 20 to page 13.

Page 13: *Mismatches occur when only one of the methods detects blowing snow, when the other does not : N $BS_{ceilo}$ if blowing snow is only reported by the BSD algorithm, but not reported by the visual observations (commission error), and N $BS_{vis}$ when only the visual observer records blowing snow, referred to as ommission error.*
and page 20
*Furthermore, the hourly time filtering applied leads to commission and ommission errors (short-lived events are likely removed from the running mean).*

**10 Question 7: p.17, line13: "Blowing ...". Sentence is unclear**

The sentence has been adapted.

*At both stations, blowing snow layers with the highest vertical extend occur during blowing snow mixed with precipitation. Mean blowing snow layer height during precipitating event reaches 331 m, while clear sky mean blowing snow layer depth is limited to 78 m at PE.*

**11 Question 8: p.20, last line. "effect of katabatic winds on blowing snow ...". Strong katabatic winds may occur in the presence of strong synoptic scale winds in Adélie Land and are associated to blowing snow. Please specify that your statement concerns PE and Neumayer.**

The sentence has been adapted accordingly.

*This, together with the reduced number of blowing snow events occurring under katabatic winds might indicate that the effect of katabatic winds on blowing snow occurrence has been overestimated at both PE and Neumayer stations, and that synoptic events bringing fresh snow is a most possibly determining factor for blowing snow at those locations.*

**REVIEWER 3 General comments:The authors have properly addressed my main concerns. There remain few very minor typos/correction to be implemented (in my opinion):**

**12    Question 1: P.5, l.7: I suggest to add "algorithm" between "BSD" and "detects".**

5    The abstract has been adapted accordingly.

*The BSD algorithm detects heavy blowing snow 36% of the time at Neumayer (2011-2015) and 13% at Princess Elisabeth station (2010-2016).*

**13    Question 2: Fig.6: the red/orange horizontal line is not described in the caption. I guess it corresponds to the**
10    **height of the blowing snow layer, but this should be mentioned (or the line removed, as it appears in Fig.S3-b)**

Yes, this line is the height of the blowing snow layer. The legend is now updated (see below).

*In that case, the range gate at which the profile increases again is the top of the blowing snow layer, and the base of the cloud and/or precipitation (the green line around the 7th bin in Fig.1, for the black and the red profiles).*

15   **14    Question 3: P.14, l.6: "attenuated" before $\beta_{att}$ should be removed.**

Indeed. the sentence has been adapted accordingly.

*We can therefore apply the BSD algorithm in the same fashion to both datasets with the only difference being the $\beta_{att}$ near surface threshold (first step in the BSD algorithm used to identify the presence of blowing snow).*

20   **15    Question 4: P.18, l.30-31: the explanation is still not clear to me, although from the responses I understand that this could be related to a kind of sampling effect(?). In the responses, the authors state that there is less probability to have longer periods without any precip, I think this should also be mentioned in the text (to help the reader).**

Indeed, the phrasing is not very clear. Very long periods without any precipitation (300 hours, more than 10 days) are rather rare at PE station, and it seems that blowing snow happened during a significant portion of these occasional measurements.

*A possible explanation is that there are less measurements as we go in time, as very long periods without any precipitation (300 hours, more than 10 days) are rather rare at PE station, and that blowing snow occurred during those measurements.*

[Figure]

**Figure 1.** Different types of hourly-averaged one-event profiles relevant for blowing snow measured by the ceilometer at PE station: blue line - typical blowing snow signal with no precipitation nor clouds (24-04-2016); red line - blowing snow overlaid by precipitation (10-02-2016); black line - precipitation in the absence of blowing snow (10-02-2014); yellow line - near-zero signal for clear sky conditions (24-04-2016). The height above ground is indicated on the right axis and the corresponding bin number on the left axis. All profiles exclude the lowermost bin, and start at the second bin (15 m agl.). The grey lines represent the discontinuity between bins 4 and 5 (35-45 m). The arrows indicate the presence of precipitation. The re-increase in ceilometer profile, indicator of the presence of clouds/precipitation is indicated in green.

**16    Question 5: P.20, l.8-9: "slightly lower than ... strongly reduced compared to average conditions"**

the sentence has been adapted accordingly.

5    *The surface pressure is slightly lower than, and the temperature inversion is strongly reduced compared to average conditions.*

**17    Question 6: Fig.12: what could explain the second peak around 50h?**

This has not been investigated. A quick analysis showed that the events leading to a blowing snow layer height exceeding 300 m are characterized by higher wind speeds than the events leading to blowing snow layers below 300 m vertical extent. In addition, the re-increase represents a few events, compared to the higher density of lower extend blowing snow layers for the

10    same time lag. Regarding the fact that between  20 and 50 h time lag, this effect is not visible, it can be linked to the 'sampling effect' and due to chance (see question 4 above).

[revised manuscript text omitted]

**1 Equation of the different metrics used in section 4.1.**

Equations come from Allouche et al. (2006),

with $a = $ N $BS_{both}$ , $b = $ N $BS_{ceilo}$, $c = $ N $BS_{vis}$, $d = $ N $BS_{none}$ and $n = a + b + c + d$.

$$accuracy = \frac{a+d}{n} \tag{1}$$

$$sensitivity = \frac{a}{a+c} \tag{2}$$

$$specificity = \frac{d}{b+d} \tag{3}$$

$$kappa = \frac{(\frac{a+d}{n}) - \frac{(a+b)(a+c)+(c+d)(d+b)}{n^2}}{1 - \frac{(a+b)(a+c)+(c+a)(d+b)}{n^2}} \tag{4}$$

$$TSS = sensitivity + specificity - 1 \tag{5}$$

[Figure]

[Figure]

**Figure S1.** Annual cycle of blowing snow frequency at PE station, for the period 2010-2017. The error bar represents the inter annual variability. Note that the measurements between June and October are present in 2015 only.

**Figure S2.** Availability of data for each of the PE instruments mentioned in the manuscript. Dark areas represent missing data, light grey areas represent available data and vertical lines represent a period when the instrument was not set up yet.

**References**

Allouche, O., Tsoar, A. and Kadmon, R.: Assessing the accuracy of species distribution models: prevalence, kappa and the true skills statistics (TSS), Journal of applied Ecology, 43, 1223–1232, 2006.

**Table S1.** Detail of the sensors and range used to measure the different meteorological variables by the AWS/IWS at PE station.

| Variable measured | sensor | range $\pm$ accuracy |
|---|---|---|
| 2m temperature | Vaisala HMP35AC | -80 to 56 °C $\pm$ 0,3 °C |
| 2m humidity | Vaisala HMP35AC | 0 to 100 % $\pm$ 2% |
| 2m wind direction | Young 05103 | 0 to 360 ° $\pm$ 3 ° |
| 2m wind speed | Young 05103 | 0 to 60 m$\cdot$s$^{-1}$ $\pm$ 0,3 m$\cdot$s$^{-1}$ |
| 2m pressure | Vaisala PTB1011B | 600 to 1060 hPa $\pm$ 4 hPa |
| 2m short wave radiation | Kipp CNR1 | 0 to 2000 W$\cdot$m$^{-2}$ $\pm$ 2% |
| 2m long wave radiation | Kipp CNR1 | -250 to 250 W$\cdot$m$^{-2}$ $\pm$ 15 W$\cdot$m$^{-2}$ |
| height of the instrument | SR50 | 0,5 to 10 m $\pm$ 0,01 m |

**Table S2.** Details of sensors used to measure the different meteorological variables at Neumayer station

| variable measured | sensor |
|---|---|
| 2 and 10 m temperature | Thies 2.1265.10.000 PT-100 platinium resistance sensors |
| 2 m dew point temperature | Vaisala HMP233 hygrometers |
| relative humidity | dew point temperature and temperature |
| surface air pressure | Digiquartz 215-AW002 |
| wind vector | Thies 4.3323.21.002 cup anemometer and wind vane combined |

**Table S3.** WMO categorization of blowing snow

| WMO code | description |
|---|---|
| 0 | snow haze |
| 1 | Drifting snow, light or moderate, with or without snow falling |
| 2 | Drifting snow, heavy, without snow falling |
| 3 | Drifting snow, heavy, with snow falling |
| 4 | Blowing snow, slight or moderate, without snow falling |
| 5 | Blowing snow, heavy, without snow falling |
| 6 | Blowing snow, slight or moderate, with snow falling |
| 7 | Blowing snow, heavy, with snow falling |
| 8 | Drifting and blowing snow, slight or moderate, impossible to determine whether sno is falling or not |
| 9 | Drifting and blowing snow, heavy, impossible to determine whether sno is falling or not |

**Table S4.** Detection rate of the different categories of observations. $BS_{both}$ stands for the part of blowing snow detected by both the algorithm and the visual observations, $BS_{none}$ is when both methods agree that there is no blowing snow. $BS_{ceilo}$ and $BS_{vis}$ represent detections rates by the algorithm and the observer only, respectively. All columns are expressed in %. B stands for blowing and D for drifting snow. The total number of measurements is 10584.

|  | $BS_{both}$ | $BS_{none}$ | $BS_{ceilo}$ | $BS_{vis}$ |
|---|---|---|---|---|
| B and D snow, with or without precipiation (WMO cat 01 to 09) | 22 | 48 | 9 | 21 |
| B and D snow, without precipitation (WMO cat 02, 04 and 05) | 9 | 61 | 22 | 8 |
| heavy B snow, without precipitation (WMO cat 05) | 3 | 70 | 27 | 0 |
| all B snow, without precipitation (WMO cat 04 and 05) | 7 | 66 | 23 | 4 |
| all B snow, with or without precipitation (WMO cat > 03) | 17 | 61 | 14 | 8 |
| heavy B snow, with or without precipitation (WMO cat 05, 07 and 09) | 10 | 67 | 21 | 2 |

[Figure]

**Figure S3.** Determination of the height of the layer by the BSD algorithm. (a) in case of a cloudless blowing snow profile, the height of the layer (thin horizontal line) is attained when the backscatter intensity reaches the clear sky threshold (dashed vertical line). (b) in case of precipitation, the height of the blowing snow layer (dashed horizontal line) is reached when the intensity of the backscatter signal re-increases.

[Figure]

**Figure S4.** Mean profiles for each of the detection categories at Neumayer: $BS_{both}$ when both methods detect blowing snow, $BS_{ceilo}$ when blowing snow is reported by the BSD algorithm only, and $BS_{vis}$ if blowing snow is detected by the visual observer, but not the BSD algorithm

**Table S5.** Meteorological conditions at Neumayer during the different events, for years 2011-2015 (mean $\pm$ standard deviation).

| variable (units) | $BS_{both}$ | $BS_{ceilo}$ | $BS_{vis}$ | $BS_{none}$ |
|---|---|---|---|---|
| number of occurences | 3416 | 23344 | 1451 | 1113 |
| temperature 10m (°C) | -14.8 $\pm$ 06 | -15.3 $\pm$ 08 | -10.5 $\pm$ 06 | -13.1 $\pm$ 08 |
| temperature 2m (°C) | -14.8 $\pm$ 06 | -15.5 $\pm$ 08 | -10.5 $\pm$ 06 | -13.2 $\pm$ 08 |
| air temperature (°C) | -14.8 $\pm$ 06 | -15.8 $\pm$ 09 | -10.5 $\pm$ 06 | -13.4 $\pm$ 08 |
| wind speed 10m (m$\cdot$s$^{-1}$) | 21.6 $\pm$ 04 | 13.3 $\pm$ 04 | 17.3 $\pm$ 03 | 11.4 $\pm$ 03 |
| wind speed 2m (m$\cdot$s$^{-1}$) | 18.2 $\pm$ 03 | 11.3 $\pm$ 03 | 14.6 $\pm$ 03 | 9.6 $\pm$ 03 |
| wind direction 10m (°to N) | 93.7 $\pm$ 30 | 118.0 $\pm$ 68 | 89.8 $\pm$ 27 | 111 $\pm$ 64 |
| wind direction 2m (°to N) | 93.7 $\pm$ 31 | 118.3 $\pm$ 68 | 89.8 $\pm$ 27 | 111.3 $\pm$ 65 |
| relative humidity (%) | 85.4 $\pm$ 04 | 81.0$\pm$ 06 | 88.3 $\pm$ 05 | 81.9 $\pm$ 08 |
| pressure (hPa) | 972.3 $\pm$ 11 | 979.3 $\pm$ 09 | 972.6 $\pm$ 09 | 978.8 $\pm$ 10 |
| dew/frost point (°C) | -16.4 $\pm$ 06 | -18.0 $\pm$ 09 | -11.8 $\pm$ 09 | -15.5 $\pm$ 09 |
| height (m) | 340.0 $\pm$ 170 | 112.0 $\pm$ 122 | - | - |

**Table S6.** Meteorological conditions at Princess Elisabeth during the different events, for years 2010-2017 (mean ± standard deviation)

| variable (units) | absence of blowing snow no precipitation | absence of blowing snow with precipitation | blowing snow no precipitation | blowing snow with precipitation | heavy mixed events |
|---|---|---|---|---|---|
| number of hours | 12 671 | 10 232 | 1032 | 948 | 1490 |
| wind direction (°to N) | 147 ± 66 | 119 ± 65 | 120 ± 65 | 107± 60 | 87 ± 47 |
| wind speed ($m \cdot s^{-1}$) | 4 ± 4 | 5 ± 3 | 7 ± 2 | 7± 4 | 10 ± 5 |
| shortwave in ($W \cdot m^{-2}$) | 194 ± 206 | 173 ± 226 | 138 ± 253 | 142± 206 | 130 ± 187 |
| shortwave out ($W \cdot m^{-2}$) | 155 ± 169 | 141 ± 182 | 116 ± 197 | 119± 172 | 112 ± 157 |
| longwave in ($W \cdot m^{-2}$) | 154 ± 46 | 197 ± 37 | 190 ± 27 | 218 ± 36 | 238 ± 36 |
| longwave out ($W \cdot m^{-2}$) | 218 ± 31 | 236 ± 33 | 234 ± 36 | 243 ± 31 | 251 ± 29 |
| air temperature (°C) | -16 ± 6 | -15 ± 6 | -16.5 ± 7 | -15 ± 6 | 14 ± 6 |
| specific humidity ($g \cdot kg^{-1}$) | 0.65 ± 0.6 | 0.9 ± 0.6 | 0.96± 0.5 | 1.2 ± 0.7 | 1.4± 0.7 |
| relative humidity (%) | 46 ± 17 | 63 ± 18 | 73 ± 16 | 80 ± 16 | 94 ± 13 |
| pressure (hPa) | 827 ± 9 | 828 ± 7 | 827 ± 8 | 828 ± 9 | 827 ± 11 |
| surface temperature (°C) | -24 ± 8 | -19 ± 9 | -20 ± 10 | -18 ± 8 | -15 ± 8 |
| temp. inversion ($°C \cdot m^{-1}$) | 8.0 ± 3 | 4.4 ± 4 | 3.5 ± 4 | 2.5 ± 3 | 1.2 ± 2 |
| temp. gradient ($°C \cdot m^{-1}$) | 2.5 ± 1 | 1.4 ± 1.0 | 1 ± 1.4 | 0.8 ± 1 | 0.5 ± 8 |
| height of blowing snow layer (m) | - | - | 78 ± 272 | 331 ± 643 | 255 ± 267 |

**Table S7.** Meteorological conditions at Neumayer III during the different events, for years 2011-2015 (mean ± standard deviation)

| variable (units) | non blowing snow without precipitation | non blowing snow with precipitation | blowing snow without precipitation | blowing snow with precipitation | heavy mixed events |
|---|---|---|---|---|---|
| number of hours | 9599 | 11 385 | 1237 | 3834 | 10 351 |
| wind direction 2m (°to N) | 191 ± 59 | 135 ± 68 | 77 ± 06 | 81 ± 06 | 65 ± 06 |
| wind speed 2m ($m \cdot s^{-1}$) | 4 ± 3 | 5 ± 3 | 6 ± 4 | 7 ± 5 | 13 ± 6 |
| air temperature 2m (°C) | -23 ± 11 | -14 ± 9 | -21 ± 10 | -12 ± 8 | -15 ± 7 |
| relative humidity (%) | 73 ± 7 | 79 ± 8 | 77 ± 6 | 81 ± 6 | 85 ± 6 |
| height of blowing snow layer (m) | - | - | 139 ± 180 | 110 ± 159 | 1303 ± 1581 |